# The Royal Zoological Society of Scotland’s Approach to Assessing and Promoting Animal Welfare in Collaboration with Universities

**DOI:** 10.3390/ani14152223

**Published:** 2024-07-31

**Authors:** Kristine M. Gandia, Jo Elliott, Simon Girling, Sharon E. Kessler, Hannah M. Buchanan-Smith

**Affiliations:** 1Psychology, Faculty of Natural Sciences, University of Stirling, Stirling FK9 4LA, UK; 2Royal Zoological Society of Scotland, 134 Corstorphine Road, Edinburgh EH12 6TS, UK

**Keywords:** animal welfare, animal welfare assessment, zoo, five domains, positive welfare

## Abstract

**Simple Summary:**

All good zoos have animal welfare assessment policies and practices in place to promote their animals’ welfare, 24/7, across the lifespan. Here, we describe the approaches that The Royal Zoological Society of Scotland’s (RZSS) Edinburgh Zoo and Highland Wildlife Park take to providing conditions for animals to thrive in captivity, in collaboration with universities. We describe the governance and philosophical stance that RZSS takes, including policies and review processes, and how they are put into practice. In collaboration with universities, a wide range of welfare research is conducted, which maps well to the Five Domains model of animal welfare and enables the zoo to incorporate the findings into their practices. Finally, we outline how the University of Stirling has collaborated with the RZSS to make its welfare assessment tool for the whole animal collection more efficient and reliable, and to provide a quantitative evidence base for individuals/groups that require in depth behavioural data to understand what interventions might be appropriate. We advocate for collaboration across zoos on behavioural projects to improve understandings of “baseline” behaviours in captivity, across the 24-h day, seasons, and life stages, as well as in relation to a range of housing and husbandry practices.

**Abstract:**

Good zoos have four aims—to conserve species, educate the public, engage in research, and provide recreation—all of which can only be achieved when underpinned by high animal welfare standards. In this paper, we share the approach that The Royal Zoological Society of Scotland’s (RZSS) Edinburgh Zoo and Highland Wildlife Park take to animal welfare. We highlight the role that animal welfare research, in collaboration with universities, has had in enabling the zoo to take an evidence-based approach to welfare and to put findings into practice. We share the collaborative process through which we developed and piloted the current animal welfare assessment tools, how they were validated, and how they were tested for reliability as part of a long-term collaboration between the Royal Zoological Society of Scotland and the University of Stirling: (1) the RZSS Welfare Assessment Tool, a 50-question animal welfare assessment adapted from the British and Irish Association of Zoos and Aquariums (BIAZA) Toolkit; and (2) the Stirling Toolkit, a package of evidence-based resources for behavioural-data collection. Our aim is to facilitate standardised, evidence-based approaches to assessing animal welfare which, when finalised, can be used collaboratively across zoos.

## 1. The Role of Zoos in Animal Welfare Science

Over the last three decades, the importance of improving welfare standards for captive animals has substantially increased [1]. This movement has been a result of the scientific community’s developing understanding of sentience and pain perception in non-human animals [2,3,4,5]. Sentience is the general capacity to feel, for instance, feeling hunger, satiety, pain, joy, boredom, and excitement; however, it does not require animals to reflect on feelings or understand what others may be feeling [6]. Advancements in understandings of sentience have also increased public concern in animal welfare across different sectors (farm, laboratory, and captive wild animals), and this concern has created political pressure and driven policy and legislative change directed at improving the care of animals [7]. However, much of the research relating welfare to sentience has been for vertebrates, because they are the majority of animals involved in animal research [2] and used as livestock [4], with the top five species researched on this topic being pigs, cows, sheep, chickens, and rats [5]. Zoos and aquariums are unique among captive animal organisations in that they house many vertebrate and invertebrate species which do not have extensive research conducted on their ability to experience pain (physical or psychological), their possession of sentience, their natural behaviours, or their physiology. However, in choosing to keep animals in captivity, zoos have ethical and moral responsibilities to uphold high standards of welfare practices [8,9]. 

Modern zoos have four main goals: conservation, education, research, and recreation [10]. All four goals are interconnected and also underpinned by the need to uphold high standards of welfare practices. To conserve species, zoos need collective support from the public for the vision of conservation and high/consistent visitor numbers to fund their conservation research and efforts. For zoos to gain public backing for conservation, visitors need to understand the reasoning for conservation through education, be invested in the survival of species, and have positive perceptions of zoos so that they are willing to visit recreationally [7,11]. Research is an important tool for many conservation efforts. It provides information on species to guide conservation efforts and also scientific evidence on the effectiveness of conservation. Having healthy animals, with an extensive natural behavioural repertoire, is fundamental to both good welfare and the success of animal conservation projects. Research into what constitutes good welfare and how it can be accurately assessed is the foundation to this process. To encourage zoo visitors to return frequently, the visitors must have positive perceptions of zoos and their goals, and particularly of animal welfare. Zoo visitors’ perceptions of zoos and animal welfare is greatly influenced by whether they witness animals displaying positive or negative behaviours [12]. 

Godinez et al. [12] conducted a study investigating the correlation between visitors’ categorisation of jaguar (*Panthera onca*) behaviour as being stereotypic or species-typical and their perception of the jaguars’ well-being, the exhibit quality, and the quality of their experience. They found that visitors who correctly categorised stereotypic behaviours as abnormal (roughly half of the visitors), generally rated the well-being of the jaguar, exhibit quality, and their own experience lower than those visitors who categorised the stereotypic behaviour as a normal active behaviour. This study aptly demonstrates how the public can be astute in recognising welfare concerns and how this would negatively affect their experience at a zoo and, very likely, their willingness to return to a zoo. The public having a negative perception of welfare and, consequently, a zoo can affect conservation efforts because it was found that a conservation opportunity was 20 times more effective on-site in a zoo than off-site to encourage conservation-related behaviour [13]. Therefore, it is likely beneficial for zoos to provide as many in situ conservation opportunities for visitors (i.e., signing petitions, donating to conservation organisations of their choice, etc.) [14] and to find ways to motivate visitors to return to the zoo often. 

Aside from welfare having an indirect effect on conservation efforts through its influence on public perception of zoos, welfare also has more direct effects on conservation. Greggor et al. [15] argue that improving welfare directly improves conservation by promoting natural behaviours that can lead to more successful reintroductions and breeding attempts in situ. They follow the “Opportunities to Thrive” principles, where animals are given the opportunity to have a strategically presented, well-balanced diet that is self-maintained for optimal health; the opportunity to express species-typical behaviour; and the opportunity to have choice and control. Greggor et al. [15] mentioned that providing naturalistic environments and diets and promoting natural behaviours will allow the animal to develop necessary survival skills for reintroduction, while equally improving welfare for the animal while in captivity. In relation to breeding, having good welfare and natural breeding opportunities that allow for the display of species-typical breeding behaviours (i.e., providing birds with nesting materials to display natural nest-building behaviours) can help zoos understand when their animals are ready to breed. In addition, giving animals the opportunity to care for offspring improves welfare for the offspring’s long-term coping ability. For the parent, it helps in avoiding the stress initiated by the removal of offspring, and it allows the parent to develop rearing skills and bonds with its offspring. These outcomes would improve breeding efforts in the long-term ex situ and in situ, should the individuals be released [16]. 

Research is therefore the tool that provides this information to guide and validate conservation practices and a means to communicate advances in conservation efforts to all organisations involved with conservation. Zoo research has also played a direct role in advancing our understanding of animal welfare and progressing welfare practices and frameworks. The welfare movement progressed from wanting to stop animal cruelty to stopping suffering, and then to preventing suffering, and now, in modern day, to preventing negative welfare and promoting positive welfare [17]. The Five Freedoms was the first highly influential welfare assessment framework that emerged to address increasing concerns of welfare for livestock focused on preventing suffering [18]. The Five Freedoms were an internationally accepted standard for the welfare of animals, stating that animals must have freedom from (1) thirst and hunger, (2) discomfort, (3) pain, injury or disease, and (4) fear and distress, and they must have (5) the freedom to express normal behaviour. The Five Freedoms model focuses on providing basic needs of survival to animals, emphasising the avoidance of negative welfare states. Though this model was highly influential for the field of animal welfare and formed the basis of legislation for the care of captive animals, it does not consider positive affective states and lacks detail on the variety of negative affective states, resulting in a model that does not provide a basis to determine the internal/external conditions and cognitive processes that would lead to those states [19]. With a better understanding of affective states and sentience in non-human animals, the field of animal welfare developed, and more emphasis was placed on providing animals opportunities for positive experiences rather than only trying to avoid negative states. 

The Five Domains model for animal welfare assessment was developed as a response to the changing views on positive affective states within the field of animal welfare. The model is a framework for identifying positive and negative states in four physical/functional domains and one mental domain [20]. The physical/functional domains include nutrition, environment, physical health, and behaviour. Importantly, the behaviour domain (now termed “behavioural interaction”) includes the influence of human interactions on welfare [21]. This is essential because zoo animals regularly interact with zoo staff and visitors. Together, the state of these four domains influences the mental domain and indicates the overall welfare status of the individual animal. The negative and positive welfare states of the four physical/functional domains are reflected as positive and negative affective states in the final mental domain. In this way, the Five Domain model emphasises the importance of sentience (ability to feel) and accounts for all the factors that may result in changes to the affective state of an animal, putting the animal at the centre of the welfare assessment. 

With these interconnected goals and practices, zoos operate as hubs for collaboration, connecting the public, conservation organisations, and field-based projects. These kinds of practices follow the highly regarded “One Plan Approach” to conservation, coined and promoted by the International Union for Conservation of Nature (IUCN) Species Survival Commission (SSC) Conservation Planning Specialist Group (CPSG) [22], and the similar One Conservation concept [23]. The One Plan Approach suggests that species conservation planning should include all populations of a species (in situ and ex situ) and be developed by all responsible parties to create management strategies and conservation actions for a single conservation plan across all populations [22,24]. Developing holistic and collaborative approaches like the One Plan Approach would address the interconnectedness of the four goals for modern zoos to conserve, educate, conduct research, and provide recreation while providing high standards of welfare care to animals. 

In this paper, we first explain the broader context of Royal Zoological Society of Scotland’s (RZSS) animal welfare procedures and policy for Edinburgh Zoo and the Highland Wildlife Park. Next, we describe the reciprocal benefits of collaboration between the RZSS and universities, including the history of in-depth welfare research collaborations across the Five Domains (nutrition, health, environment, behavioural interactions, and mental state). This collaboration often enables RZSS to put welfare science insights into practice. Finally, we describe the formal standardised practices for assessing the welfare of the whole animal collection. We describe the collaborative process through which we developed and piloted the current animal welfare assessment tools, how they were validated, and how they were tested for reliability as part of a long-term collaboration between the Royal Zoological Society of Scotland and the University of Stirling. First, there is the RZSS Welfare Assessment Tool, a 50-question animal welfare assessment adapted from the British and Irish Association of Zoos and Aquariums (BIAZA) Toolkit, which is based on the Five Domains approach; and, second, there is the Stirling Toolkit, a package of evidence-based resources that facilitate taking an evidence-based approach to behaviour and environment questions.

## 2. Welfare Practices at the Royal Zoological Society of Scotland (RZSS)

The Royal Zoological Society of Scotland is a wildlife conservation charity with a bold vision: a world where nature is protected, valued, and loved [25]. RZSS operates both Edinburgh Zoo and the Highland Wildlife Park (near Aviemore). As of July 2024, Edinburgh Zoo houses over 130 species and more than 4000 animals (around 3500 being invertebrates). Highland Wildlife Park houses 31 species, with more than 3300 animals (nearly 3000 being invertebrates). Of the invertebrates, most are partula, pine hoverfly, and dark-bordered beauty moths, with close to 2000 planned to be reintroduced by the end of the year. The RZSS strategy to 2030 includes three pledges: to reverse the decline of at least 50 species by 2030, to create deeper connections with nature for more than a million people by 2030, and to enable more than 100 communities to better protect nature by 2030. It is not possible to achieve any of these strategic aims without first ensuring that the animal collections at Edinburgh Zoo and the Highland Wildlife Park are provided with opportunities for the best possible welfare. The animals in these collections serve as ambassadors for their species, for the habitats in which their wild counterparts live, and for RZSS’s conservation efforts in Scotland and beyond [26].

The welfare of the zoo-housed animals is promoted daily by the dedicated keeper teams. Keepers are all trained to a high standard, in line with the European Professional Zookeeper Qualification Framework [27]. It is the keepers who prepare and deliver the nutritious diets the animals need; ensure that they have clean water readily available; and ensure that their environment is safe, secure, clean, and within the parameters expected. Keepers are also trained to recognise and respond to both physical and behavioural changes in their animals which may be indicative of poor health or welfare.

The design of exhibits is key to ensuring the animals we keep can be provided with opportunities to display a wide range of natural behaviours in an appropriate social group. The exhibits are enhanced with environmental enrichment to increase variability and provide challenges for the animals, thus providing opportunities to show agency [21]. An in-house team of tradespeople and gardeners is on hand to maintain and repair exhibits as required. 

More widely, RZSS has an on-site veterinary team led by zoo and wildlife veterinary specialists, who provide both preventative and reactive veterinary care for our animals, and who work with the keepers to design appropriate diets. We have an on-site fully equipped registered veterinary treatment facility and pharmacy, and off-show space to house animals which require some quiet space to recover from illness or to undergo quarantine. RZSS has a Population Management Policy which covers ethical animal acquisition and disposition; decision-making regarding hand rearing; decision-making regarding management euthanasia; and other means of controlling surplus animal stock, e.g., contraception.

RZSS also has an external and independent welfare and ethics committee (Animal Welfare Advisory Group) that reviews the policies and procedures affecting the animals kept by them and any significant welfare challenges that occur. The Animal Welfare Advisory Group has Terms of Reference which require a wide range of expertise, including animal welfare, veterinary, zoo biology or captive-animal management, conservation, field research, and ethics. At least one member of the Board of Trustees and two members of the keeping staff also attend as full members of the advisory group. The group meets three times per year, rotating between Edinburgh Zoo and the Highland Wildlife Park. After the meeting, members walk around the zoo to observe animals, learn about the ongoing research, and see enclosures and exhibits where changes are being enacted/considered.

## 3. The History of Research on Animal Welfare at RZSS in Collaboration with Universities

Universities have a key role in society by offering educational and training opportunities, conducting research, and collaborating with local organisations and governments to keep abreast with changing perspectives and to seek solutions to challenges. Long-standing relationships between RZSS and universities have enabled RZSS to embed animal science, including welfare, into the governing structures and lead to reciprocal benefits for both the zoo and the field of animal welfare science. For example, academics sit on the RZSS Board of Trustees, and their Animal Welfare Advisory Group (see [28]), advising on up-to-date thinking in animal welfare science and changing ethical principles. RZSS has a committed Discovery and Learning team in Edinburgh, and a Wildlife Discovery Centre opened in 2024 at the Highland Wildlife Park. Education is delivered from nursery to higher education and includes additional needs’ support and outreach programs [29]. Enthusing others about animals and their conservation and promoting their welfare are important, as highlighted in the RZSS strategic plan 2017–2022 strategy tagline “From empathy to action” [30]. Research has also assessed how visitors engage with the interpretation and educational materials, evidencing where efforts should focus to engage visitors with education messaging [31]. 

Zoos provide unique experiential training opportunities, given the breadth of animals they house. In linking with universities, academics may bring students to the zoo as part of their degree courses to learn methodologies for collecting behavioural data and may target key topics for which having many observers simultaneously may be an asset (e.g., [32]). Such zoo-based practical experiences are an excellent method of teaching. Student evaluations consistently emphasise that the students would much rather be involved in original research conducted in the zoo that provide useful results that have implications for the management and promotion of welfare of captive animals than simpler practical work used to demonstrate the concepts and methodological issues. Similarly, research conducted by academics contributes to online learning packs for teachers and summer-school activities with RZSS (e.g., [33]). 

An additional benefit to people from university collaboration is understanding job satisfaction and occupational stressors and suggesting ways to ameliorate the stressors [34]. Whilst for many zoo professionals, caring for animals is described as a calling, a survey (*n* = 311 zoo professional in a range of roles) indicated that they were not able to do their best for animal welfare and indicated that their own welfare and their perception of the welfare of animals in their care were linked [34]. Such studies are useful to understand organisational aspects within zoos that, with careful management, may lead to policies and practices that reduce stressors experienced by staff; increase satisfaction in their job, potentially with improved staff retention; and, critically, may allow staff to work at their best to promote animal welfare. 

Whilst the dedicated staff at RZSS is focused on taking care of their animals, often they require skilled researchers with specialised software and equipment and familiar with the range of methodologies and analyses available to assess animal welfare. Welfare assessment measures range from judgement bias, facial expressions, vocalisations, behaviour, social network analyses, physiology, and thermal imaging. Understanding how to interpret these measures in relation to welfare valence is critical and RZSS has contributed to these discussions, including whether comparing the behaviour of animals in zoos with their wild conspecifics is a valid welfare indicator, using the giraffe (*Giraffa camelopardalis*) as a model [35]. In the next section, we summarise a selection of the many in-depth animal welfare projects that have taken place at RZSS that tap into the key elements of the Five Domains model [21], which underpins the welfare strategies of the major Zoo Associations (e.g., World Association of Zoos and Aquariums (WAZA), Association of Zoos and Aquariums, European Association of Zoos and Aquaria (EAZA), and BIAZA).

### 3.1. Examples of the Breadth of Collaborative Welfare Research Projects Conducted at RZSS

The Five Domains model (Figure 1; [21]) is the most accepted framework for animal welfare assessment. The four physical/functional domains (nutrition, physical environment, health, and behavioural interaction) all feed into associated affects—and it is this affective state that determines the animal’s welfare. Responding to requests from RZSS staff, researchers often follow up issues on which keepers are keen to receive in-depth data and analyses, or researchers choose their own topics. Where keepers are key to the research, they are rightfully included as authors (e.g., [36,37,38]), and veterinary staff regularly conduct independent research on health issues and publish new methodologies (e.g., [39,40,41,42,43,44]). 

Appendix A provides a summary of selected studies conducted at RZSS since 1996 and the key findings in relation to animal welfare. Accurate welfare assessment and interventions are at the heart of any process, and good zoos are now incorporating welfare evaluation processes into their core responsibilities, not just when keepers and visitors are present, but considering the 24/7 approach across the whole of an animal’s life to incorporate circadian and circannual rhythms and different life stages, with concomitant needs [45]. A recent example of such a study on circadian and circannual rhythms at RZSS was on giant pandas [46,47].

The first of the five domains is nutrition—the correct intake of food and water, in relation to the quantity, quality, and variety. If nutrition does not meet the animal’s needs, then examples of the associated affects are thirst/water intoxication, hunger/starvation, malaise due to malnutrition, eating-related boredom, feeling bloated, gastrointestinal pain, and nausea. Optimal nutrition leads to positive effects of pleasures of drinking, food tastes, smells, mastication, and gastrointestinal comfort and satiety [21]. In addition to ongoing monitoring of the quantity and quality of animal diets, nutritional value, and food presentation, research at RZSS has explored some behavioural effects of diet, including its effect on regurgitation and reingestion in chimpanzees [48]. Although more appropriately categorised under the behavioural interaction (fourth) domain, it is worth noting that RZSS performs research on a wide range of feeding-enrichment techniques designed to promote natural foraging behaviour and prevent boredom (e.g., bush dogs [49], felids [50,51], and parrots [52]). The importance of food presentation has clear consequences for both animal welfare and conservation [53].

The second domain relates to physical-environmental conditions. The negative conditions that are associated with this domain are, for example, overcrowding; inappropriate substrates; pollutants; aversive noise or odours; thermal extremes; and unpredictable events that may lead to affects such as discomfort, anxiety, and exhaustion if restful sleep is not possible. The goal is to provide environments that lead to physical, thermal, olfactory, auditory, and visual comfort and make sure that the animals are well rested and experience with congenial variety and predictability. Research conducted at RZSS that has tackled this domain includes studies on space [54,55]; animal responses to noise/music [56,57]; properties of objects in relation to control and complexity [58]; predictability [59]; and nesting, sleeping, and nighttime behaviours [60].

The health domain is concerned with all aspects of an animal’s welfare that are related to disease, injury, and different levels of physical fitness [21]. There are a range of affective qualities that are related to this domain, including many expressions of pain (e.g., debility and nausea); effects of being too fat or too thin, with associated metabolic consequences; and poor physical fitness, leading to muscle weakness and exhaustion. The goal is to provide environments that, for example, prevent disease; minimise functional impairment; and allow animals to experience the comfort of good health, functional capacity, and vitality of fitness. At RZSS, studies have focused on health assessments, including in animals destined for reintroduction (beavers [61]), the use of positive reinforcement training to assist with veterinary treatments [37], methodological contributions (e.g., [43]), and disease description and factors affecting spread [40,41]. The review on the tiger feeding poles is a prime example of how increasing physical activity has significant positive impacts on the skeleton [62].

The fourth domain is behavioural interaction, and it has three subcategories: (a) interaction with the environment, (b) with other non-human animals, and (c) with humans. Here “agency” is the key term—“when animals engage in voluntary, self-generated and/or goal-directed behaviours” ([21], p. 13), and when enclosures are well designed and with plentiful environmental enrichment, with appropriate social groups, the animals have opportunities to show agency. In interactions with the environment, when agency is impeded in barren, confined environments with inescapable sensory imposition and lack of choice, animals may be bored, depressed, frustrated, hypervigilant, etc. When agency is promoted, animals are more likely to feel calm, in control, energised, and focussed [21]. RZSS is committed to providing naturalist environments that promote these positive effects, and research has focussed on space use, choice, and control (e.g., [48,63,64,65]). 

In the second subcategory—interactions with other animals include those of both the same and other species. Housing animals in appropriate social groups where they can choose to affiliate or avoid other enclosure occupants is key, and when compatible conspecifics are housed together in suitable environments, there may be opportunities for affectionate sociality, maternal/paternal or group rewards in rearing young, playfulness, sexual gratification, and security. Research at the RZSS has found that personality plays a key role in the quality of relationships between animals [66] and that methods of measuring personality [67] and using social network analyses are useful tools for measuring and understanding group dynamics [68]. RZSS also houses many animals in mixed-species groups, many of which would naturally associate in the wild (e.g., primates [38,69] and penguins [70]), increasing the complexity and providing stimulation for the animals. In all social situations, careful monitoring is important during introductions [71], and as individuals age and change (e.g., in dominance) within groups [72]. Understanding space use (e.g., [73]; Daoudi-Simison et al. submitted) and modifying the environment to minimise competition over key resources are critical (e.g., [38]). 

The final key subcategory is interactions with humans, considering their attitudes, voice, aptitude, and handling/controlling management, which may lead to the negative effects of escape/avoidance, cowering, anxiety, fear, and persistent unease or to positive effects, such as being at ease, being compliantly responsive, being calm, and being confident [21]. Research at RZSS includes that on positive-reinforcement training (storks [74]; chimpanzee [37]). The visitor effect in zoos is well studied and may positively or negatively impact on animal welfare depending on a range of factors (e.g., [75]). Research at RSZZ has investigated visitor impacts. For example, no effect of visitors was found on the rate of regurgitation and reingestion in chimpanzees [48]. A study exploring visitor density and visitor eating behaviour found that primates spent more time looking at visitors who were eating [32]. This research is important, as it may affect the placement of picnic sites.

### 3.2. Welfare Research within Public Engagement with Science Research Centres at RZSS

RZSS Edinburgh Zoo has two public engagements with science centres: Living Links to Human Evolution Research Centre (hereafter Living Links), housing mixed species groups of South American capuchin (*Sapajus apella*) and squirrel monkeys (*Saimiri sciurius*); and the Budongo Research Unit within the Budongo Trail exhibit, housing chimpanzees (*Pan troglodytes*). Living Links and Budongo Trail are purpose-built facilities dedicated to providing world-class housing and husbandry, as well as for collaboration with universities for research purposes. Both research units receive core funding from the University of St Andrews and provide research opportunities to the Scottish Primate Research Group, a consortium of Scottish Universities and beyond. The key aspects of their design that make them public engagement centres—not just research centres—are that they are set up so that the public can watch studies as they are conducted. Both facilities have testing rooms with transparent walls that the public can see into. This enables the public to learn about both the findings and the process of the science that is being conducted on the primates. 

In both the research centres, the welfare implications of engaging in research are carefully monitored, and policies are in place to ensure researchers are fully trained, and any negative welfare consequences are avoided [76]. The primates participate in the research on an entirely voluntary basis, entering a research area and leaving the area and returning to the group when they indicate to the researcher that they wish to do so [64,76,77]. The researchers use a variety of methods to record the behaviour of the primates, including monitoring eye movements [78] and measuring how their surface temperature varies, using thermal imaging [79], as doing so can indicate arousal and welfare state. The findings on how the primates learn, communicate, and interact with their social and physical worlds (e.g., [80]) are communicated to the visitors. This understanding of animal behaviour and cognitive capacities is known to affect the desire of people to connect with animals, aiding conservation efforts (e.g., [81]). 

Both centres have benefitted from links to wild populations and provide connections for visitors to see how the RZSS is supporting conservation initiatives. Budongo Trail is linked to the Budongo Conservation Field Station (BCFS) in Uganda helping protect chimpanzees and other forest cohabitants in the wild, promoting education messaging about threats facing wildlife (see [82]), and working closely with the BCFS to promote welfare in wild chimpanzees, for example by removing snares. Living Links was formally opened in 2008 [76]. The Living Links enclosure design and positioning of substrates has also been informed by studies in the wild (e.g., in Suriname, [83]) to encourage natural locomotor patterns, and vertical separation of the two species. These wild links encourage the implementation of in-depth knowledge of the habitats, behaviours, and cognitive challenges primates face in the wild to be incorporated into a naturalistic captive environment that promotes welfare. For example, wild chimpanzees engage in fission–fusion, where, depending on resources such as food, they may split up into smaller groups (fission), and then join up again (fusion) into larger groups [84]. The Budongo keeper team managed this process over a period of 15 months and won the BIAZA Bronze Behaviour and Welfare Award, recognising their efforts in this new management system to improve behaviour and integration within the troop. In addition, the RZSS chimpanzees are part of the European Endangered Species Programme (EEP), overseeing the genetics of managing the wider chimpanzee population to promote health and diversity.

The research centres are also key to training future generations by the RZSS Discovery centre team, for providing remote learning resources (see [85]) and more recently promoting Citizen Science [86]. At Living Links, visitors are trained to collect and classify data using established standardised techniques, contributing to our longstanding project about monkey welfare and enclosure use [72,73]. It is hoped that these opportunities will improve awareness of the scientific process and increase visitor curiosity, potentially converting visitors to volunteers to collect data using the Stirling Toolkit: Evidence-Based Resources to support RZSS in their decision-making concerning animal welfare.

## 4. Collaboration between RZSS Edinburgh and University of Stirling to Develop and Pilot Welfare Assessment Methods

RZSS Edinburgh Zoo and RZSS Highland Wildlife Park are both accredited members of BIAZA, EAZA, and WAZA. The WAZA 2023 Animal Welfare Goal states, “WAZA regional associations must have an animal welfare evaluation process in place”, and “all WAZA institutional members must be compliant with this process” [87]. In this section, we describe how the RZSS welfare assessment process was developed and piloted. It is compliant with a component of the WAZA 2023 Animal Welfare Goal.

We highlight the ongoing collaborative process through which we validated the RZSS welfare assessment process and increased its efficiency and reliability. Our collaboration has two main outputs: (1) the RZSS Welfare Assessment Tool, which is a 50-question welfare assessment covering the Five Domains; and (2) the Stirling Toolkit: Evidence-Based Resources, which are resources that are part of an evidence-based protocol for collecting data on the behavioural and environmental questions in the RZSS Welfare Assessment Tool. The following sections describe (Section 4.1) the challenges that must be overcome in order for an assessment process to be standardised, evidenced-based, and practical for whole zoo assessments; (Section 4.2) the refinement of the RZSS Welfare Assessment Tool, including streamlining it for efficiency and evaluating its validity; and (Section 4.3) the development and pilot of the Stirling Toolkit: Evidence-Based Resources, demonstrating the reliability and feasibility of applying behavioural data to the assessments. The process of piloting and finalising is ongoing. The RZSS animal welfare tool and Stirling Toolkit can be used together or separately as best suits the needs of the zoo/facility.

### 4.1. Challenges of Whole Zoo Formal Welfare Assessments

Many good zoos are eager to comply with the WAZA 2023 animal welfare goal of conducting regular welfare assessments and want to follow suggestions on welfare assessments from more local associations, like EAZA and BIAZA. However, many zoos around the world are facing similar difficulties while implementing welfare assessment programs. Because assessments of welfare must cover the full range of biological and mental needs of the animal, they require much time, many resources, and trained staff to conduct them. To begin with, the shift in prioritisation of welfare means that management priorities must also be shifted, which can be a slow process. In addition, since welfare is a relatively new concept, the knowledge of what welfare is and the resources available for learning about and conducting welfare assessments are still limited. Then, there are the more practical issues, like zoos housing a diverse number of species with many varying needs and limited research on their life history and behaviour. This makes it difficult to have validated and reliable welfare indicators for all species. The number of species and individuals also makes it difficult to fit welfare assessments into the regular work routine. Then, collating that information from the many staff and departments involved is another difficulty. Once they have collected the information, a final difficulty is to interpret the data to implement effective interventions if needed. Through collaboration, RZSS Edinburgh Zoo and the University of Stirling are trying to address these difficulties through the development of a welfare assessment tool and evidence-based welfare toolkit. The components of the tool and toolkit are being developed to make the welfare assessment process reliable, efficient, and able to provide clear and effective outcomes. 

Jones et al. [88] compared several prominent welfare assessment frameworks and tools to advise on a process to develop efficient, reliable, and validated welfare assessment tools. To develop the RZSS Welfare Assessment Tool and Stirling Toolkit, we followed this process suggested by Jones et al. [88] (Figure 2). 

The first step outlined in the Jones et al. [88] process is choosing a framework to structure the assessment (Figure 2). This step was completed by JE prior to the start of this collaboration in March 2021. JE created the initial RZSS Welfare Assessment Tool by adapting the BIAZA welfare toolkit, which itself follows the Five Domains approach to welfare assessment [17,21]. However, for our collaboration, we decided we wanted to incorporate the strengths and minimise the weaknesses of the several prominent and popular welfare frameworks/processes available today. 

Though we focused on the benefits and highly accepted structure of the Five Domains framework, other prominent frameworks also have many strengths and benefits. These frameworks include the Universal Animal Welfare Framework [89], the Animal Welfare Assessment Grid [90], and the European Welfare Quality^®^ project [91]. However, one important aspect of welfare that none of these frameworks address is the continuous nature of welfare. These frameworks do not apply the cyclical nature of the day, week, and seasons to the approach of assessing welfare, nor do they address how welfare may be affected through different life stages. A framework that has been proposed, which aims to address this, combining principles from the Welfare Quality^®^ project and Five Domains, is the 24/7 across the lifespan approach to welfare assessment [45]. This approach emphasises having a holistic method to welfare that considers the natural history of an animal and how the context of a zoo environment affects natural cycles. The authors are clear it is not an animal welfare assessment though. The approach aims to map out and research whether the needs and wants of captive animals are being met 24/7, across the lifespan. Though some frameworks encourage that rhythms be considered for aspects of welfare like the changing diet composition throughout the year, none goes as in-depth to suggest that welfare is continuous on multiple time scales and should be assessed as such. This approach to welfare assessment is therefore unique and thorough in that it takes an evolutionary perspective to welfare assessment. 

To complete the first step of the Jones et al. [88] welfare assessment development process of deciding on a framework, we considered the strengths and weaknesses of prominent frameworks to try and incorporate all of their strengths and minimise their weaknesses. Table 1 summarises the strengths and weaknesses of popular and prominent frameworks/processes for welfare assessment. Ultimately, the strengths we wanted to incorporate were a balance between resource-based and animal-based indicators of welfare; an assessment of positive and negative animal indicators; an evidence-based approach to understanding animal outputs in the context of their environment and in response to changing external factors, sustainability, and efficiency, without being labour- and resource-intensive or compromising reliability; the inclusion of assessments by care staff familiar with the animals; and addressing cycles and the continuous nature of welfare (a few of these also being suggested for consideration under the “develop tool” and “pilot tool” steps by Jones et al. [88], Figure 2).

In the following subsections, we describe the “develop tool” and “pilot tool” steps (Figure 2) for the development of the RZSS Welfare Assessment Tool and the Stirling Toolkit: Evidence-Based Resources. This will be followed by how we plan to return to the “Development” step and make changes based on initial piloting, to then pilot again before the tool and toolkit are finalised.

### 4.2. Refining the RZSS Welfare Assessment Tool

Before the collaboration with University of Stirling, RZSS had established a welfare assessment tool that was based on the BIAZA tool. The collaboration began as a way to continue the development of the RZSS Welfare Assessment Tool to make it more reliable and efficient. The initial welfare assessment tool consisted of 50 questions, split into five categories: (1) nutrition, 4 questions; (2) health, 9 questions; (3) environment, 14 questions; (4) husbandry, 9 questions; and (5) behaviour, 14 questions. These questions could be responded to with four possibilities: yes, no, unknown, or not applicable. The RZSS process consisted of three or four assessors, with each responding to all 50 questions. Assessors for each assessment were a combination of the curator, species keepers, animal team leaders, and veterinarians but did not always include an assessor from each of these groups. The behaviour category was completed by conducting a single 15-min observation session and then responding to questions. The keeper would also use previous knowledge and awareness of their animals’ behaviours to respond to questions in this category. Once all assessors had responded to all 50 questions, the assessors would have a meeting to come to a consensus on the responses. These outcomes are incorporated and compiled into the Welfare Action Plan, which is reviewed by the Animal Welfare Advisory Group (Section 2). The Welfare Action Plan is a complete listing of all welfare challenges recorded from the welfare assessments across the collection. For each challenge recorded, there is a description of the challenge and its severity on a five-point scale. The scale includes (A) no challenge but listed for completeness; (B) low; (C) mild–moderate; (D) marked–severe; and (E) very severe—requires immediate action. As each challenge is addressed, it is tracked through listings as: no action required, in progress, pending, monitored, controlled, or resolved. Approval for the proposed solution may include a range of stakeholders depending on the level of the welfare challenge encountered, the need for an urgent mitigation, and the resources required for the proposed solution. These decision-makers may include anyone from the animal section team leaders to the CEO depending on the severity, urgency, and potential cost of the proposed solutions. 

To determine how we wanted to improve the RZSS Welfare Assessment Tool, we first needed a consultation between the collaborative team to ensure the changes would be fit for purpose. In our consultation in March 2021, we were all in agreement that the questions covered all the key resource-based (input) factors that affect welfare, as well as key animal-based (output) factors, and would be effective in accurately determining the welfare state of animals. However, the assessment was not validated, reliability of responses was unknown, and the process was inefficient with the various assessors each responding to the entire assessment. Therefore, although it was a thorough assessment process, we wanted evidence on its validity and reliability and to determine if the number of assessors could be reduced to make the process more efficient.

#### 4.2.1. Methods: Testing Validity and Reliability, and Improving Efficiency of the RZSS Welfare Assessment Tool

To test the validity and reliability of the RZSS Welfare Assessment Tool, we conducted an analysis on the responses from 17 assessments completed for individuals and groups from several species (Table 2). These species were the chimpanzee (*Pan troglodytes*), giant panda (*Ailuropoda melanoleuca*), Malayan tapir (*Tapirus indicus*), mandrill (*Mandrillus sphinx*), eastern white pelican (*Pelecanus onocrotalus*), sulphur-crested cockatoo (*Cacatua galerita*), Argentine giant tegu (*Salvator merianae*), yellow-breasted capuchin (*Sapajus xanthosternos*), Amur tiger (*Panthera tigris altaica*), greater one-horned rhinoceros (*Rhinoceros unicornis*), and domestic yak (*Bos grunniens*). To begin the improvement of the welfare assessment tool, we wanted to first determine whether the questions themselves were valid. We had to establish that the individual questions within each category were properly testing the overarching question of each category. Since, in the assessment, a “yes” response was indicative of a positive welfare indicator and a “no” response of a negative welfare indicator, we would expect covariance of responses within each category if the individual questions were all addressing the overarching question accurately. We tested this with Cronbach’s alpha for responses across assessments but within each category, as well as for the overall assessment (all 50 questions). Cronbach’s alpha is appropriate because it gives a measure of covariance relative to variance within these categories. A higher Cronbach’s alpha is desired, as it is indicative of covariance within these categories. In addition to measuring the consistency within each category, we also conducted an omitted variable analysis where the Cronbach’s alpha is recalculated when each question is omitted one at a time. When the Cronbach’s alpha increases after omission, this suggests that the omitted variable tends to have a response that has high variance within the category and does not consistently covary with the other responses within the category. These results could pinpoint specific questions that need to be modified to improve validity.

The next step in improving the welfare assessment tool is to determine the reliability of responses and also whether the number of assessors could be reduced for certain questions. As of July 2024, the RZSS Edinburgh Zoo and Highland Wildlife Park, together, housed approx. 150 species and approx. 7700 animals (nearly 6500 being invertebrates). With all assessors responding to all 50 questions, including questions pertaining to categories outside of their expertise (e.g., veterinarians answering questions on husbandry), the process of assessing all animals could take years. Therefore, we wanted to determine whether it would be appropriate to have assessors focus their efforts in areas of their expertise: veterinarians answering health questions; keepers answering husbandry, behaviour, and environment questions; and the curator collecting information for the nutrition category, along with bringing general expertise to the husbandry, behaviour, and environment categories. We first tested the reliability of responses with Fleiss Kappa, which determines the concordance between assessors’ responses for each individual animal/group assessment. We analysed the agreement between assessors for the 17 individual animals/groups (Table 2). The concordance between assessors could then indicate whether it would be appropriate to reduce the number of assessors for each question. For instance, if there was high agreement between all assessors on all questions, one assessor could reliably provide representative responses to all questions. If there is disagreement between assessors for certain questions, then those questions would likely be kept to multiple assessors to encourage multiple perspectives and discussion.

#### 4.2.2. Results: Validity of Questions and Categories in Welfare Assessment Tool

Within each category of the assessment, there seemed to be decent concordance (Table 3), suggesting that the covariance within each category was high and the categories were likely addressing the overarching theme well. It is important to note that, for questions assessing animal outputs, assessors evaluating a group are instructed to respond with “no” if the statement does not apply to at least one individual in the group (Appendix A). The environment, husbandry, and behaviour categories had good internal consistency (0.89 ≥ α ≥ 0.80) [95]. The nutrition category had an acceptable amount of internal consistency (0.79 ≥ α ≥ 0.70), and the health category had a questionable amount (0.69 ≥ α ≥ 0.60). No categories had poor (0.59 ≥ α ≥ 0.50) or unacceptable (α < 0.59) internal consistency. As a whole, the assessment had excellent reliability (α ≥ 0.90). The omitted variable analysis revealed that 7 questions of the 50 in the assessment seem to cause variance within their respective categories (Table 4). 

For environment, the omitted variable analysis suggested that the environment questions that seemed to have lower covariance with the rest of the category’s questions were exclusively questions that asked about whether different environmental conditions (temperature, humidity, and light) were appropriate year-round. These are factors are difficult for zoos, in general, to measure year-round if a specific system to regulate these factors throughout the day and seasons is not in place. Because of this, a response to this question becomes more subjective. For instance, in a single assessment, an assessor might score “yes” if no issues were encountered throughout the year, but another might score “unknown” because there were no measurements, and a third might score “no” because the species had to be housed indoors for some days of the year due to more extreme weather conditions. The remaining questions in the environment category do not ask about the appropriateness of conditions year-round. Therefore, these three questions causing variance in the environment category highlight an issue that many zoos face in not having the resources to monitor environments year-round. 

For husbandry, the omitted variable analysis revealed that the question on whether the water feature is safe and allowed for species/specific behaviours caused the most variance within the category. The difference was minimal, and the reason it likely did not covary with the other responses was that, for 50% of assessments, it was not applicable. However, to improve the reliability of the response to this question in general, the data collected from the enclosure use can be used to determine how often they are around the water feature and what behaviours they display when they are around the water feature.

For the behaviour category, the only question which increased the Cronbach’s alpha when omitted was on whether play behaviour is displayed. This is a rare behaviour, particularly with adult animals, and, therefore, it may likely cause variance within the category because it is unlikely for the assessors which are not the animal keepers to catch it in the 15 min observation session performed for the assessment. The keepers, who know their animals, can recall having seen them play or not, while other assessors would only have the ability to respond to the question from the 15 min observation. This behaviour is unlike mating/reproductive behaviours, the other behaviours specifically asked about in the assessment, because zoo staff do monitor mating/reproductive behaviours closely, and all relevant staff would be notified when these behaviours are displayed outside of the welfare assessment. This result of the omitted variable analysis showing play as causing variance highlights an area where the behavioural-data collection would be useful and increase reliability of this response since assessors can reference evidence from more extensive behavioural observations to respond to it.

The two categories with lower, but still acceptable, Cronbach’s alpha values were nutrition and health. For nutrition, the omitted variable analysis revealed that the question on whether a diet sheet was available caused the most variance within the category. However, it should be noted that, since this category had only four questions, it is easier for one question to cause variance. Regardless, this question coming out in the omitted variable analysis shows how having assessors respond to questions in their area of expertise may perhaps be helpful. The welfare curator is responsible for collecting information from the literature on best-practice guidelines for diet and nutrition and sharing that information with keepers. Therefore, their response to this question would be the most valuable and accurate.

For health, the question on whether faeces were healthy seemed to cause the most variance within the category. The improvement of the reliability and consistency of the responses to this question could also be achieved through using the Stirling Toolkit for data collection. This could be a survey that is conducted periodically on the formation of faeces that has images of normal and abnormal faeces formation for the species. This kind of survey would help to avoid different interpretations of what normal faeces formation is and create a record of faeces formation. The data record could then be accessed at the time of the welfare assessment to respond to this question more accurately and consistently.

Overall, there was good consistency and covariance within the different categories of the assessment with only a few questions that caused some variance. These data can help to target what areas of the assessment can be improved with an evidence-based approach to produce more consistent responses or questions that would be better addressed by the staff with the most accurate information. Sherwen et al. [94] similarly produced a welfare assessment that assessed both resource inputs and animal outputs based on the Five Domains. However, the internal consistency of each category was not measured. These data demonstrate a method of validating assessments and highlighting areas that can be expanded or improved.

#### 4.2.3. Results: Reliability and Efficiency of Assessor Responses

The average Fleiss Kappa value for agreement between assessors across assessments was 0.3483 (Table 5), which is a fair amount of agreement (0.40 ≥ K ≥ 0.21) between assessors, but on the lower half of the Kappa statistic scale [96]. To understand the origins of this agreement/disagreement between assessors, we also analysed the Fleiss Kappa values for each possible response in the assessment (Table 6). Each question had four possible responses: no, yes, not applicable, or unknown. The only response with substantial agreement (0.80 ≥ K ≥ 0.61) was “not applicable” (averaged K = 0.6662). The main responses of “yes” (averaged K = 0.3829) and “no” (averaged K = 0.2649) both had fair agreement (0.40 ≥ K ≥ 0.21). The “unknown” (averaged K = 0.0925) response had slight agreement (0.20 ≥ K ≥ 0.00).

These results indicate that most of the agreement between assessors came from questions which were not applicable to the individual/group. This could potentially skew the agreement to be higher due to responses that are not based on the animal. However, the poor agreement between unknown and blank responses could also skew the agreement to be lower. 

The substantial disagreement on the unknown and blank answers suggests that when one assessor wrote unknown or left the response blank, there would be another assessor which did respond with yes or no. These results support the argument that it may be more effective to have assessors respond to questions where they have relevant or specialised knowledge, without losing reliability of accurately identifying the agreed response. This process may look like having veterinarians answer health-category questions or questions pertaining to physical health, keepers answering questions on behaviour and husbandry, and the welfare curator answering questions that require searching through the literature for best-practice guidelines or species requirements. In this manner, there would be a smaller workload for each assessor, while maintaining the accuracy of responses to each category of questions. This would allow for a more efficient welfare assessment process, which is already resource-intensive, and would mean more assessments can be completed in shorter timespans. This suggestion can be further justified by the fact that, after all assessors complete the assessment, they have a consultation to create a consolidated assessment for that individual animal or group that the welfare interventions are then based on. Therefore, for questions which are left blank or unknown for certain assessors, the consolidated assessment would still agree with the assessor who did provide a response.

The averaged agreement for the “yes” (averaged K = 0.3829) and “no” (averaged K = 0.2649) responses was similar to the overall agreement between assessors across all assessments, which is a good sign that the assessors have decent agreement on definitive answers that is reflected across all assessments. However, the slight disparity in agreement between “yes” and “no” responses may suggest that there is more agreement on when an animal is in good condition than when it is not. In the assessment, a “yes” response was indicative of a positive welfare state, and a “no” response of a negative welfare state. Therefore, these results may indicate that it is more difficult for assessors to identify and agree on a welfare concern. This seems to be the case more for individual assessments, where the agreement for “no” (averaged K = 0.1878) was slight (0.20 ≥ K ≥ 0.00), than for group assessments, where the agreement for “no” (averaged K= 0.3517) was fair (0.40 ≥ K ≥ 0.21). This difference in “no” responses between individual and group assessments should be investigated further, with a larger number of assessments and a greater variety of species to determine if it is due to the kinds of species assessed or truly a difference between individual and group assessments.

Given these results, although we do suggest that the number of questions each assessor answers is reduced by only answering questions they are specialised to answer, some questions may overlap in expertise between assessors and may still need multiple assessors to ensure welfare concerns are accurately spotted. The next step in making the assessment more efficient would be to determine which questions should be answered by specialised staff only and which questions overlap between assessors and would still benefit from multiple assessors.

### 4.3. Development and Pilot of Stirling Toolkit: Evidence-Based Resources

Animal outputs are the only measures which can indicate the affective state of an animal [97]. Often, however, welfare assessments are limited on the evidence-base for animal outputs [98]. The initial RZSS Welfare Assessment Tool used a 15-min observation session to respond to behaviour questions. Though a reasonable estimate may be gained from a 15-min observation session, systematic data collection accounting for more full cycles of behaviour (i.e., daytime, nighttime, and seasonality) would help in adding more context to the observed behaviours and, consequently, result in more informative conclusions. In these instances, the keepers’ long-term knowledge of displayed behaviours by the animals in their care can be incredibly helpful in directing data collectors toward behaviours to focus on, but relying entirely on keeper’s recall could result in some inaccuracies due to the keepers having many animals in their care and a variety of responsibilities. In addition, several environment questions ask if certain environmental features are appropriate, but as in many zoos, evidence is not always collected on whether the resource inputs are achieving their intended purpose with the animal. 

The Stirling Toolkit was developed to empower zoos to take an evidence-based approach to the environment and behaviour domains by systematically collecting feasible amounts of behavioural data. In addition, we want to encourage and facilitate zoos to begin collecting data on cycles of behaviour. Many aspects of an animal’s welfare state (hormonal cycles, sleep quality, breeding, metabolic processes, and general physiological health) are influenced or regulated by daily, annual, and life cycles [99,100,101,102]. Therefore, incorporating the assessment of behavioural rhythms solves the large concerns with using the expression of natural behaviours as an indication of positive welfare and the potential misinterpretation of time budgets [93]. Rhythms can provide the context that activity budgets ignore by identifying whether behaviours are expressed in the appropriate context and at times of day, year, or life stage that are adaptive. Therefore, this allows us to more accurately interpret neutral behaviours as positive or negative. In addition, we want to address a major drawback with many of the frameworks (Table 1)—the disconnection between the assessment of the resource inputs and animal outputs. Our evidence-based approach will look at behaviours in response to their environment and toward specific environmental features through enclosure use data. These data allow us to directly connect the resource inputs to animal outputs by displaying whether the desired animal outputs are achieved in response to specific environmental features and not simply assumed. This approach of investigating natural rhythms in conjunction with their response to the environment would also allow us to understand the contexts in which positive welfare states are achieved so that conditions can be mimicked and positive welfare promoted. Conversely, the detailed context in which negative welfare states occur can also be discerned, resulting in more targeted and reliable interventions being proposed to more directly and effectively address welfare issues. 

In this section, we describe the development and piloting of the Stirling Toolkit: Evidence-Based Resources. The toolkit is being developed to be a package of resources which will enable users to collect and analyse behavioural data (ethograms for use on the *ZooMonitor* App [103], quizzes testing knowledge of those ethograms, and training videos for using *ZooMonitor*). We describe how we selected species for piloting the Stirling Toolkit, developed ethograms, piloted data collection with a team of university students, and successfully applied the data they collected to the environment and behavioural domains. This demonstrates proof-of-concept and feasibility for the Stirling Toolkit.

#### 4.3.1. Species Selection

To determine whether the evidence-based approach to the behaviour questions is effective for assessing welfare across the species held in the RZSS Edinburgh Zoo collection, we chose species spanning across vertebrate taxa (ongoing piloting is adding invertebrates). We also ensured that species which have limited information on their natural histories and behaviour are represented in the sample, thus avoiding biasing the assessment toward species which have had extensive research on their behaviours and natural histories. This ensured that the assessment could properly determine the welfare of species even when background information is limited, as is true for many species in zoo collections. In addition, choice of species was also determined by those which had a recent welfare assessment at the time of data collection, or an assessment that would occur simultaneously with behavioural data collection. With these criteria, we selected 10 species across four of the major taxa in vertebrates (fish, mammals, reptiles, and birds) (Table 7). These species were the yellow-breasted capuchin (*Sapajus xanthosternos*), brown capuchin (*Sapajus apella*), chimpanzee (*Pan troglodytes*), meerkat (*Suricata suricatta*), Nubian giraffe (*Giraffa camelopardalis camelopardalis*), eastern white pelican (*Pelecanus onocrotalus*), anemone fish (*Amphiprion ocellaris*), corn snake (*Pantherophis guttatus*), Taiwan beauty snake (*Orthriophis taeniurus*), and milk snake (*Lampropeltis triangulum*). 

Whether the species was assessed at the individual or group level was determined by RZSS Edinburgh Zoo and affected by ease of individual identification. This allowed proof-of-concept for whether evidence could be reliably collected on the individual and group level, depending on the needs of the zoo. To ensure that group- and individual-level observations were capturing the same data, we compared results between individual and group observations for the pelicans. 

#### 4.3.2. Ethogram Development and Corresponding *ZooMonitor* Projects

In total, there were 19 questions that could be answered with systematic behavioural observations. These questions are listed in Table 8 and Table 9. This includes all questions from the behaviour category and five questions from the environment category. In creating the ethograms, we ensured that each question could be answered with direct observations from the ethogram. Overall, the 14 behaviour questions are designed to ask about positive and negative behaviours and, in general, species-typical behaviours. Ethograms within each taxon had general behaviours that would apply to most species within that taxon and then additional behaviours which might be more species-specific. For instance, locomotion, feeding/foraging, and resting/sleeping were behaviours added to all mammal ethograms, but a behaviour like burrowing would be specific to the meerkat ethogram. In addition, all ethograms had behaviours for capturing interactions with visitors (positive: attention on or interaction with a visitor; negative: aggressive or agonistic toward visitor; did not include ignoring a visitor) and keepers (attention on a keeper; engaging with direct feeding or training session), as well as enrichment items. Therefore, we created skeletal ethograms within each taxon that could then be modified according to the species. We added behaviours or categories that were specific to welfare concerns that keepers may have flagged for their species and wanted more detailed data on. For instance, keepers might wish to know how alert an animal was of a predator which was housed nearby and in its line of sight. 

We used the online application *ZooMonitor* to conduct our observations. The ethograms were converted into *ZooMonitor* projects. For observations on individuals, focal sampling was used with interval and all occurrence sampling for relevant event behaviours. For group projects, interval scan sampling was used. The session lengths and number of intervals can be seen in Table 10. 

The flexibility of the app allowed us to additionally address five questions in the environment domain which normally cannot be answered in high detail with traditional behavioural ethograms. These questions, in general, ask if the environmental features are appropriate, allow for species-specific behaviours, and are useful across different climatic conditions (Table 9). *ZooMonitor* has a useful feature where heat maps of the animal’s location within the enclosure can be created. Enclosure maps were uploaded to the app that were either aerial views of the enclosure or renditions that indicated where key environmental features were. With this, we were able to track how often the animals were in specific areas of their enclosure and how much of the enclosure was actually used by the animal. In addition, heat maps could be created for specific behaviours from the individual observations. Therefore, we could determine where in the enclosure the animals were more likely to display specific behaviours, like resting/sleeping or locomotion. We could also see enclosure use on days with specific weather conditions from both individual and group observations by adding questions at the start of the session on the weather conditions and if they were housed outdoors or with access to the outdoors. All of these features allow for an evidence-based approach to the welfare assessment questions on whether the environment is appropriate and whether resource inputs are allowing for positive animal outputs. This would not be possible without the flexibility and user-friendly interface of *ZooMonitor*.

#### 4.3.3. Sustainability of Data Collection and Analysis through Collaboration with University Students

One of the main goals of this pilot was to evaluate the feasibility of creating a self-sustaining protocol to measure animal outputs continuously. Zoos have difficulty maintaining these kinds of programs because of the limited resources with staff and time [45]. We contributed by creating a program with the University of Stirling, where students who are completing their theses, placements, or simply volunteering could collect behavioural data on animal outputs. Collaborating with university students to collect and analyse data avoids having to require keepers to collect these data and for welfare staff to perform the bulk of the analysis. The time commitment for those engaging in behavioural data collection can be found in Table 11. Training students with different levels of experience and background in animal behaviour demonstrated proof-of-concept that people with little/no experience in behavioural data collection with *ZooMonitor* can successfully participate. This approach can later be extended to the general public through vetting of volunteers since *ZooMonitor* is user-friendly for non-experts. Though we did not train non-university students in this study, overtime, a range of projects with varying levels of difficulty can be made for continuous behavioural monitoring that would allow people with different skill levels to collect behavioural data.

#### 4.3.4. Reliability Testing

In total, six observers assisted in data collection across the different species. Students were writing dissertations or completing placements and were therefore assigned to one to three species to observe. Observers were trained on the ethograms and the use of *ZooMonitor* in person (Table 11). To produce data that would be used for the welfare assessment, observers had to pass reliability testing. Since testing reliability purely from live observations results in many ethogram behaviours not being evaluated [104], we designed our reliability testing with two stages, aiming to cover all ethogram behaviours as best as possible. The first stage was an ethogram quiz for which the observer had to receive >80%. The quizzes covered the definitions of each behaviour within the ethogram with a mixture of multiple choice, true/false, fill in the blank, definition, and scenario questions. 

The second stage was live inter-observer reliability using *ZooMonitor* in person at the zoo. In the future, this stage can be completed with pre-recorded videos. Regardless of whether the data collection was for group or individual assessments, the observer would perform reliability with an individual animal. The observer had to match 70% of observations with the lead investigator (KMG) to pass. Observers were not allowed to begin data collection until they passed two or three in-person reliability tests, depending on the difficulty of the ethogram. For simpler ethograms, like the pelican ethogram, three sessions had to be reliable. Observers were allowed unlimited attempts to pass, though reliability testing was ended if too many consecutive sessions were failed. In the future, when volunteers are incorporated, this same cut-off can be applied as to not require so much effort and time from zoo employees who are training volunteers. 

#### 4.3.5. Data Collection

Once reliability testing was passed, observers were allowed to collect data for use in the welfare assessments. Data collection was scheduled so that sessions were spread across the day, with the final spread of data being 2–4 sessions per hour during opening hours (10am to 4 or 5pm depending on the season). This systematic data collection enabled us to gain an estimate of the daily cycles of behaviour and enclosure use, while avoiding the bias that would occur if all sessions were recorded at one time of day. We spread the sessions across two weeks minimum to capture different weather conditions. The lengths of sessions per species and number of intervals per session can be seen in Table 10. Though the sessions are relatively short, conducting observations systematically throughout the day to be representative can provide decent estimates of behavioural states and cycles [105] (Appendix A). In the future, we would want to extend this data collection to the night, possibly employing cameras so that sessions can be completed by reviewing nighttime footage during working hours. This would allow an even more informative dataset that can be slowly built into baseline data that would emphasise sleeping behaviour as a crucial aspect of welfare. 

#### 4.3.6. Results: Piloting Results of the Striling Toolkit: Evidence-Based Resources

##### Creating a Sustainable Observation Protocol with University Students

Overall, there was success in having several students collect reliable data on a range of animals without significant addition of work to the zoo staff. The curator (JE) was consulted at the start to decide which species needed to be observed and to coordinate between researchers and keepers for each species. The keepers provided context, like housing-rotation schedules. The researchers were responsible for collecting and analysing the data. This process of collaborating with university students is a proof-of-concept of suggestions that university research can greatly improve efforts to create knowledge bases and investigate in more detail welfare risks [94].

Zoos planning to extend this to a volunteer program would need to understand the time commitment and have a system in place ready to handle the initial higher workload in establishing the program. The data-collection coordinator (the advisor at a university, researcher in a zoo, or welfare curator in a zoo) will have the responsibility of coordinating the observers, training them in behavioural data collection, and conducting reliability testing for them. To reduce the workload on one person, these steps can be managed between a team. The process of training spanned two to three weeks depending on the availability of students (Table 11). The most efficient way to complete training is to coordinate cohorts of observers to go through training together to avoid repeating the process for multiple observers individually. Each observer should go through at least two days of in-zoo practice to familiarise himself/herself with *ZooMonitor* and his/her study species. In these two days, the data-collection coordinator should be working closely with the observers to provide feedback to ensure that they are correctly identifying behaviours from the ethograms. This reduces the number of training days and the trials needed to pass reliability. 

In the future, reliability testing could be made more efficient for the data-collection coordinator, with reliability testing occurring through pre-set videos. Two types of videos can be made for reliability testing: videos which are compilations of clips of behaviours covering the entire behavioural repertoire of the taxa and videos mimicking an observation session. These videos can be given to an observer to complete remotely, reducing the effort and time invested by the data-collection coordinator. Importantly, it would also ensure that the entire behavioural repertoire of the taxa is tested for reliability, which is usually not possible with in-person, live reliability testing sessions [104]. If the reliability stage is made remote, then the training stage would not be as burdensome since it would only require coordinating the cohort for two days together in the zoo. 

In our piloting, not all students were able to pass reliability testing, meaning that their data were not useable by the zoo for welfare assessments. Therefore, procedures may need to be put in place where observers are switched to easier species if reliability is not reached after a certain number of sessions, or observers could be removed from the observation program. With university students, the latter may not be possible, as the students may be required to produce theses or reports. However, these students are still able to use their data for their course-required theses/reports and were still able to gain experience in analysing these kinds of data. However, another solution would be to establish a vetting procedure before beginning the process with observers to ensure that they are likely to be able to complete the process. 

Overall, more empirical data were collected on the species than otherwise would have been possible with only the zoo’s resources. Our results on training students with little to no background on behavioural observations and the species observed mimicked results for reliably training non-specialist students with varying levels of knowledge on elephant behaviour to reliably record behaviour [106]. This reinforces the notion that non-specialist students and, potentially, volunteers can reliably collect data and greatly increase efforts to take an evidence-based approach to assessing welfare and understanding behaviour in zoos. In the future, as the program develops, certain steps could be made more efficient by having banks of reliability videos that would only need to be distributed.

##### Incorporating Behavioural Rhythms for Greater and More Reliable Context of Animal-Based Indicators

An important aspect of the behavioural data collected is how such data address the rhythmicity of these behaviours throughout the day. Up to now, proposed frameworks for welfare assessments that evaluate animal outputs [15,91,94] only measure whether animals are displaying behaviours at all or evaluating time budgets of behaviours. However, analysing behaviour in this manner removes the context of whether behaviours are displayed at appropriate times of day or year in biologically healthy patterns. The timing and cycles of behaviour are adaptive, and just assessing time budgets or whether certain behaviours are displayed or not may be misleading. For instance, in a time budget, you may see that rest is occurring for the amount of time you would expect for the species, and this would be scored as positive. However, if the animal is resting during times of the day that are not natural, then this would actually be an indication of negative welfare which would not be captured by just time budgets. Certain questions in the assessment ask if behaviours are being displayed at the appropriate levels. These questions could be answered with time budgets, but in the following section, we demonstrate how addressing these questions with rhythms allows for more detailed conclusions to be drawn about social dynamics, interactions with the environment, and individual or group tendencies.

##### Addressing the 19 Behavioural and Environmental Questions with Evidence: Examples with Individuals and Groups

To demonstrate how a behavioural protocol could be applied to species across taxa, we conducted behavioural observations for several species (Table 10). The same welfare assessment (Appendix A) was completed for each species at either the group or individual level. The data from the *ZooMonitor* projects were used to answer all the behavioural category questions and some of the environmental category questions (Table 8 and Table 9, respectively; 19 questions total). Reports could then be created for each group or individual to assist in responding to the select 19 questions with figures generated either directly from *ZooMonitor* or within Excel from the downloaded data. 

The remainder of this section demonstrates how data from *ZooMonitor* can be presented to respond to the 19 questions by providing example data for one behaviour question, one environment question, and finally one specific welfare question not asked in the welfare assessment. The behaviour and environment examples each include two species and one individual and one group assessment to show how questions can be answered at the individual or group level. This aspect of the data collection is particularly important to demonstrate since many zoos would not have the time to conduct in-depth assessments of all individual animals but may need to conduct group assessments that could then lead to further, more meticulous assessments for any flagged individuals. From here, headings indicate which overarching question is being addressed and which species are used as an example.


*Is a negative behaviour being displayed? Snakes (individual) and anemone fish (individual)*


Similar to whether play behaviour is displayed, there is a question on whether any abnormal behaviour is observed (Figure 3). In this figure, we show how this kind of question of whether a behaviour is displayed or not (without asking about appropriate levels) can be answered with time budgets supplemented by maps to add important and informative context. In Figure 3A, we see the proportion of time the three snakes spent in different postures. We also would have recorded other abnormal behaviours, like interaction with transparent barrier, but these were not observed. However, with the snakes, we show that it is not only explicit behaviours that can be abnormal, but other species-specific aspects to behaviour as well. The rectilinear posture is when a snake takes the shape of the edges of its enclosure, indicating restriction by the enclosure. Two of three snakes assumed this position at times, which may mean that the size of the enclosure may not be appropriate. Despite the corn snake not showing this behaviour, the proportion of time spent in a coiled position (nearly 100%) may also be abnormal and a cause for concern. The map shows that the snake was only ever observed in its two shelters (bowl and box) and was never seen outside of these hiding places, hence why it was always in a coiled position. These data can lead to further investigation as to why the corn snake only stays in its shelters. For the anemone fish (Figure 3B), we see that the only individual who displays abnormal behaviour is the sharing male, which shares an anemone with the breeding pair. With the heat map showing the location where the bobbing occurs, we see that it occurs exclusively just under the anemone. This might indicate that the fish is chased out of the anemone and displays this abnormal behaviour as close to the anemone as possible. These data are a good example of how enclosure data can be applied to questions outside of environmental questions to provide context if we want to understand the potential triggers for abnormal behaviours, allowing for targeted interventions to be tested.


*Are there appropriate shelters? Brown capuchins (individual) and chimpanzees (group)*


For both the chimpanzees and capuchins, the enclosure use is the same under cloudy and sunny conditions. However, for chimpanzees in rainy conditions (Figure 4A) and capuchins in windy with no rain conditions (Figure 4B), they seem to take shelter in their indoor enclosures. Notably, the chimpanzees only seem to use one third of their outdoor enclosure, and mainly the area where food is tossed from a balcony. Therefore, for the chimps, staff may want to further investigate patterns in outdoor enclosure usage across different conditions. This may include investigating enclosure use by specific individuals, during feeding events, during specific temperatures, etc. Whether climatic conditions are appropriate for certain species living outside of their natural ecosystems and latitudes is at the forefront of welfare questions across zoos. Because if the general temperature or temperature cycles are an issue, the space can ultimately be entirely useless regardless of the presence of appropriate features in outdoor enclosures. This kind of detailed data collection facilitates examining how these separate factors synchronously affect behaviour, linking the animal outputs to environmental inputs. This helps identify the causes of welfare concerns while also determining the external factors that may promote positive welfare states. With more detailed data, welfare interventions put in place based on the evidence are more likely to be effective.


*Specific welfare question on the social dynamics of giraffe bulls housed together.*


The final example is with giraffes (Figure 5). In this situation, 5 giraffe bulls were being housed socially and keepers wanted evidence of the social dynamics and social hierarchy that was being established between the bulls. The *ZooMonitor* projects are designed to specifically answer the questions in the RZSS Welfare Assessment Tool on behaviour and environment. Therefore, this example is to demonstrate how small modifications to these projects enable staff to explore more detailed and specific welfare questions. In this case, a social modifier was added to social behaviours like follow, submissive yield, necking, fighting, and proximity where the observer would indicate which individual giraffe the focal giraffe was directing a social behaviour toward. These data allowed for a dominance matrix and sociogram (Figure 5) to be produced. However, it is important to note that, as with any project that is performed on individuals, extra training is needed for the observer to be able to distinguish between individuals. The results allowed for a tentative dominance hierarchy to be determined: Arrow -> Fenn -> Gerry -> Ronnie -> Gilbert. In addition, it was determined which individuals were affiliated most. The brothers Arrow and Ronnie tended to affiliate often, as did Fenn and Gerry. These results demonstrate the flexibility and power in the welfare ethogram projects and their potential for exploring more detailed questions or potentially related research questions without the need for extra data collection. In addition, these data were collected, and the results were generated by a Master’s placement student. This practice demonstrates the mutual benefits of engaging with universities in welfare research where students are able to develop their research skills with meaningful projects and the zoos gain insightful information on their animals.

These examples are meant to highlight the flexibility in using *ZooMonitor* and the amount of detail that can be drawn from these kinds of projects. The outcomes demonstrate how utilising this evidence-based approach can provide information for a general welfare assessment, but also to investigate in more detail individuals/groups that have been flagged for more specific welfare risks. 

### 4.4. Future Plans and Ongoing Development of the RZSS Welfare Assessment Tool and Stirling Toolkit: Evidence-Based Resources

Section 4.2 and Section 4.3 describe the process and results of the initial validation and piloting of the RZSS Welfare Assessment Tool and Stirling Toolkit: Evidence-Based Resources for Behaviour and Environment Questions. This process is ongoing, with validation and piloting continuing for the welfare assessment tool and evidence-based resources. The outcomes of the validation for the RZSS Welfare Assessment Tool demonstrate that certain questions in the assessment cause variance that might be due to ambiguity in wording or lack of evidence-based responses and low concordance between assessors may be due to assessors limiting responses to areas relevant to their knowledge base. Therefore, one of the next steps for the RZSS Welfare Assessment Tool is to hold discussions to determine how the questions that cause variance can be made more valid, either by rewording them to be less ambiguous or suggesting that an evidence-based response be required. Once the modifications to the questions are made, then we can conduct new Cronbach’s alpha tests on newly completed assessments to determine if the same questions are no longer causing variance in their respective categories. 

The next step for the RZSS Welfare Assessment Tool would be to follow up on the results from the assessor concordance. We will determine which questions tend to have agreement and disagreement between the different types of assessors (i.e., the keeper tends to disagree with the curator in question 6, all assessors tend to agree in question 27, and disagreement is common but random for question 14). If all assessors tend to agree on a question, then we would assume that only one assessor (likely the one with the most expertise) would be necessary to respond to that question. Conversely, if assessors tend to disagree in a question, then it would be suggested that several assessors respond to the question to encourage discussion. This type of delegation would reduce the number of questions each assessor would respond to, while still maintaining reliability. We would then determine if the delegation has improved efficiency of the process. These two next steps of modifying the questions that cause variance and delegating questions are currently happening, and the finalisation of the RZSS Welfare Assessment Tool should occur once these two steps are completed. The finalised RZSS Welfare Assessment Tool will guide zoos on how to prioritise welfare assessments and provide a standardised and validated assessment for implementation across a collection and even between zoos.

The piloting results for the Stirling Toolkit: Evidence-based resources showed that behavioural observations can be reliably conducted by observers with little-to-no experience, that the assessment works at the individual and group level, and that taking an evidence-based approach to behaviour and environment questions provides detail and context that would not be possible without systematic quantitative data. Development and piloting for the toolkit are ongoing. In one of the next piloting steps, we want to compare the concordance between assessors for the responses to the 19 behaviour and environment questions when assessors use only the 15-min observation session (and keeper knowledge) to when they respond using the systematically collected data with *ZooMonitor*. If we were to find higher concordance between assessors when using an evidence-based approach, it would help support our idea of using an evidence-based approach as a way to increase reliability of welfare assessments.

The finalised Stirling Toolkit is meant to be a bank of resources for taking an evidence-based approach to behavioural and environmental welfare assessment. Extra resources are currently being developed that will result in a cohesive suite of resources: generalised ethograms that cover all taxa housed in zoos, including invertebrates, corresponding *ZooMonitor* projects for the ethograms, Power BI projects for *ZooMonitor* data visualisation, reliability testing materials (quizzes and videos for testing reliability), and training materials for welfare assessments and the different software programs in the toolkit. Each of these resources is meant to address the issues zoos face in implementing welfare assessment programs and taking an evidence-based approach. The ethograms and corresponding *ZooMonitor* projects provide ready-to-use ethograms with reliable welfare indicators for behaviour and environmental interactions. Power BI projects can be used to automatically visualise data, eliminating the need for specialised staff who know how to analyse data and make visuals for data. Reliability quizzes and videos, which will include clips of all behaviours in the ethogram, will allow for remote reliability testing, reducing training staff who need to be present for reliability testing. Training materials have already been developed for the use of *ZooMonitor* and for training on how to conduct behavioural observations for welfare assessments. More training materials are being developed for the various software used in the tool and toolkit and in accordance with the gaps in training highlighted by stakeholders.

We believe that these improvements to the RZSS Welfare Assessment Tool and Stirling Toolkit will address some major difficulties zoos face in implementing a comprehensive welfare assessment program. To ensure that we are accounting for the needs of zoos, we consulted with stakeholders in zoos (welfare researchers, curators, keepers, etc.) to understand their needs and the areas where the tool and toolkit could be improved. In our current stage, we are expanding our piloting and working closely with several local zoos to further develop the tool and toolkit. Our hope is that this collaborative effort to further pilot the tool and toolkit will help us to ensure that the resources can be used across different institutions with varying needs. For instance, larger zoos may already have systems and protocols developed for welfare assessment but could benefit from the use of the Stirling Toolkit to investigate welfare concerns in more detail. Smaller zoos might not have the resources to use the Stirling Toolkit but could benefit from a ready-to-use and validated welfare assessment tool like the RZSS Welfare Assessment Tool. In addition, we hope that further development of the tool and toolkit and discussions with stakeholders can help us to make the tool and toolkit useable across all captive-animal industries, including agricultural, rehabilitation centres, and potentially laboratories.

The RZSS Welfare Assessment Tool and Stirling Toolkit are meant to be flexible in their use, providing zoos the opportunity to choose which components fit into their existing welfare programs, or to provide a complete welfare assessment package to implement. The Stirling Toolkit, in particular, is also meant to facilitate collaboration between zoos by providing standardised ethograms and data-visualisation projects to conduct behavioural research and create baselines of behaviour for less studied species to make welfare assessments more reliable and robust. Making both tools available to all captive animal organisations provides a standardised welfare assessment program that would enable zoos to compare welfare across institutions and within species around the world, allowing the zoo community to collaborate in the improvement of welfare efforts. 

### 4.5. Validity and Reliability of Welfare Assessments Applied to Zoo Animals

Reviews and articles covering the several welfare assessments available consistently emphasise the importance of having assessment tools that have their indicators validated and the reliability of the process assessed [1,88,98]. Our piloting began the process of validating the tool in several ways: (1) determining the validity of the assessment questions across several taxa, (2) determining the inter-observer reliability for assessment questions, (3) determining the reliability of animal indicators across several taxa, and (4) determining the feasibility of the entire process. Although many welfare assessment tools have been produced in the last decade following trends of placing increasing importance on animal welfare, most have not been validated, nor has their reliability been tested. This includes several welfare assessment tools and resources that are provided by zoo associations like EAZA within a welfare library.

The most validated assessments are the Welfare Quality^®^ project and the Animal Welfare Assessment Grid, which, like our assessment tool, evaluate resource inputs and animal outputs. The Welfare Quality^®^ assessment has been tested for observer reliability in several domestic animals, including pigs [107,108] and dairy goats [109]. For the pigs, inter-observer reliability was determined for a qualitative behaviour assessment (20 adjectives, like “calm” or “tense”, used to evaluate positive/negative emotions) and an assessment based on behavioural observations (30 min of observations, with scan sampling every 2 min) [107]. The results suggested that the qualitative behavioural analysis did not produce good inter-observer reliability, but the behaviour observations did. These findings support the shift toward pushing for an evidence-based approach to welfare assessment [17,98,110]. In addition, it aligns with the next steps in our validation process to assess whether there is better inter-observer reliability in responses to the behaviour and environment questions when using the evidence-based protocol versus when assessors use only the one 15-min observation session and recall.

The Animal Welfare Assessment Grid is a more evidence-based approach to welfare assessment. A thorough study determining the validity and reliability of the Animal Welfare Assessment Grid in assessing dog welfare found that it had good test–retest reliability, inter-rater reliability, construct validity, and content validity [111]. However, an important limitation to consider with these studies validating the Welfare Quality^®^ assessment and Animal Welfare Assessment Grid is that they were performed with domestic animals, as the original assessments were intended for agricultural and research animals, respectively. In addition, even within domestic species, the Welfare Quality^®^ was considered to raise concerns for validity and reliability when applied to broilers [112]. Melfi [98] highlights that many welfare indicators which are validated with domestic animals that are not closely related to the exotic animals housed in zoos may not be reliable when applied to zoo species. The Welfare Quality^®^ assessment has been successfully adapted for use on bottle-nose dolphins [113] and Dorcas gazelle [114] and proposed for the application to pygmy blue-tongue skinks [115]. Though these studies used validated animal indicators, they still need further tests to determine their reliability. And again, a limitation is that these measures are being assessed only for specific species, so they are not necessarily generalised to the vast array of species in zoos.

In contrast, the Animal Welfare Assessment Grid was specifically adapted for use in zoos [90]. However, it was not tested for reliability or validity until it was applied to gorillas [116], where it was determined to be feasible and reliable, with good agreement between researcher and keeper assessors. But the score sheets used included expected values which may have influenced scores and reduced objectivity. The more compelling test of reliability and validity for the Animal Welfare Assessment Grid in zoos was conducted in South Korea across 16 zoos [117]. This study evaluated reliability across 11 species (birds, reptiles, and mammals) in 16 zoos, with each species being scored by a researcher, veterinarian, and zookeeper. They found good inter-observer reliability, and the process was very feasible, being completed in 14 days across all zoos. This validity and reliability analysis was most similar to the process that we conducted, evaluating reliability across several taxa with multiple assessors with different expertise. Our validation and reliability testing process will be continued as we finalise the tool and pilot it across several zoos in different countries. Though the piloting completed on our assessment to date is limited—with further piloting planned to continue—our assessment is still one of the few assessments which has had its validity and reliability tested when compared to the many which are offered to zoos for adaptation. In addition, the strength in the modified RZSS Welfare Assessment Tool and evidence-based approach compared to the Animal Welfare Assessment Grid is that it includes many more indicators of positive welfare, with the evidence-based approach intended to be applied in a systematic way that allows for capturing rhythms of behaviour across the day, week, seasons, and life stages.

### 4.6. Creating Baselines of Behavioural Cycles for Later Comparison

One of the main difficulties for creating valid and reliable zoo animal welfare assessments is the lack of literature on validated animal-based indicators for the broad number of species housed in zoos [1,88,97,118]. As discussed, assessments with an evidence-based approach to animal indicators tend to be more reliable than those taking a more subjective approach [97,98,107]. However, the issue still remains that baselines to gauge the quantitative data against are uncommon due to the limited research on the biology and behaviour of many species. This means that, even from quantitative data, it can be difficult to assess welfare. 

The other main and related issue is that the zoo environment can make it difficult to determine how healthy behaviours are displayed because of the effects of the environment on behaviour and the adaptability of animals to different environments [35,119]. Howell and Cheyne [119] suggest that multiple environmental variables and animal indicators of welfare should be measured to account for behaviour patterns in a captive setting. The RZSS Welfare Assessment Tool and Stirling Toolkit are designed to do exactly this. One of the strengths of this assessment is that it links the environment and the animal and provides evidence on how the environment may affect behaviours. For instance, as was seen with the abnormal behaviour in the anemone fish and the location in which it was performed, limited space availability was suggested as the potential cause of the abnormal behaviour (Figure 3). This is unlike other welfare assessment tools that are designed to assess the environment and animal-based indicators separately [90,91]. The finalised tool will be designed to be flexible to be used to complete welfare assessments, but also to conduct research creating baselines on behavioural cycles and the environmental factors that may regulate or influence these cycles. In this manner, the welfare assessment process is self-improving because, as more data are gathered, a better understanding of each species’ biology, life history, and healthy expression of natural behaviours in a zoo context is developed. 

Although we did not measure nighttime behaviour in this study, it would be ideal for the next steps in development of this protocol to include nighttime behaviour. Nocturnal behaviour would provide details that are often overlooked in behaviour and welfare assessments [45] but that are equally important to understand and consider for welfare. Excluding nighttime activity is removing a large piece of the puzzle that influences the welfare of a species. Sleep is an evolutionarily necessary state, and assessing sleep can provide insight into sleep quality, sleeping-site appropriateness, and even social dynamics [120,121,122,123], which would all contribute to the overall welfare state of an individual. However, other behaviours apart from sleep are also important to consider when investigating nighttime behaviours. For the corn snake, activity budgets and enclosure-use heat maps from data collected between 10am and 3pm indicated that the snake was hiding in its shelters 100% of the time. This could indicate a welfare concern if the snake is mostly inactive and not making use of its entire enclosure. However, corn snakes are crepuscular. Therefore, to definitively conclude whether this is a welfare concern, we would have to record behaviour at dawn and dusk to determine if the snake is more active and uses more of its enclosure during the hours we would expect. Ignoring nighttime activities, therefore, removes much of the context that would allow us to properly monitor an animal’s welfare state. 

Given the importance of accounting for full behavioural cycles, when our welfare assessment tool and evidence-based protocol is finalised, the suggestion will be to collect data systematically to build complete baselines of behavioural cycles. For our initial piloting, our systematic approach involved conducting observations over a few weeks that eventually accumulated into an even spread of observations across the hours of the working day. This process can be expanded to collect data that would provide estimates of behavioural cycles over a full diel cycle, through each season, throughout a year, and eventually throughout the lifespan. For instance, if data collection occurs for 4 days every season, then a session can be completed every hour for 6 h, shifting that 6-h window each day until one 24-h estimate is established. This process can be repeated over the years, shifting by one month each year, which would result in one full-year estimate of behavioural cycles within 4 years. 

With this method, zoos can build baselines of behavioural cycles that would be both individual and species-specific, accounting for day, week, seasonal, and life stages and, thus, making it a powerful tool for comparison when welfare assessments occur. Individual and group data can be used long after an individual has died or a group has changed to further understand what conditions seem healthy and normal in different contexts. In addition to baselines for individuals and species within a zoo, baselines for species worldwide can be created through collaboration between multiple institutions adopting the same program and sharing ethograms. In this manner, the baselines can be built even quicker, and even more context can be understood about how the environment, different husbandry practices, and zoo location may affect behaviour and welfare. 

In addition, given that welfare assessments are meant to assess all aspects of welfare, collecting and analysing samples of a variety of hormones throughout full diel and annual cycles can be beneficial in monitoring the physiological states of animals and supplementing behavioural observations. Hormones can provide insight into metabolic health, sleep health, and mating capability or readiness [124,125,126]. Creating baselines of hormonal rhythms within a zoo setting would be immensely helpful to detect risks quickly by comparing the levels in a point in time to the expected levels for that life stage (i.e., weaning and maternal life stages), time of year, or time of day. Other external cues which would be important to determine the cycles of because they may regulate circadian rhythms are food availability [101,127], zoo events [128,129], and lunar cycles [130]. Creating baselines of different cycles and determining the effects of external cues on these cycles can help to begin answering questions on what good welfare looks like in a zoo context. These are questions at the forefront of welfare science that would be more easily addressed if a standardised welfare assessment were established that could easily be picked up by institutions worldwide. 

Having baselines for comparison would improve the validity of welfare assessment outcomes. They would also make the process more efficient by allowing the comparison of behavioural data collected by building up behavioural reference cycles for staff to compare their animal to at specific time points, similar to how a typical growth trajectory is assessed. This use of baselines also makes the welfare assessment tool and toolkit flexible in enabling certain categories, like environment and behaviour, to be assessed regularly, with just timepoints being compared, or more in depth, with a full assessment of the complete circadian and circannual rhythm of particular behaviours and the effects of environmental factors through those time periods.

## 5. Conclusions

Promoting animal welfare within zoos requires a multifaceted approach, including appropriate governance and philosophical stance, staff training, veterinary facilities, and preventive care, as well as focused research. Welfare assessments are complex, yet fundamental for achieving the conservation, education, research, and recreational goals of zoos. It is our hope that this paper and the materials we are developing will facilitate increasing collaboration across universities, zoos, and other captive-animal industries. Such collaborations have the potential to enable welfare science to answer questions that were previously difficult to address. By enabling larger sample sizes and the ability to statistically control for previously confounding variables like husbandry or geographic differences, multi-institution studies can advance welfare science by, for example, building baseline behavioural cycles of understudied species, systematically studying the effects of moving species out of the climatic and latitudinal ranges that they evolved for, examining and predicting how different species may respond to climate change, etc. As resources are developed, they will be disseminated at https://animalwelfareassessment.stir.ac.uk (accessed on 13 July 2024). We also welcome potential collaborators to contact us. 

## Figures and Tables

**Figure 1 animals-14-02223-f001:**
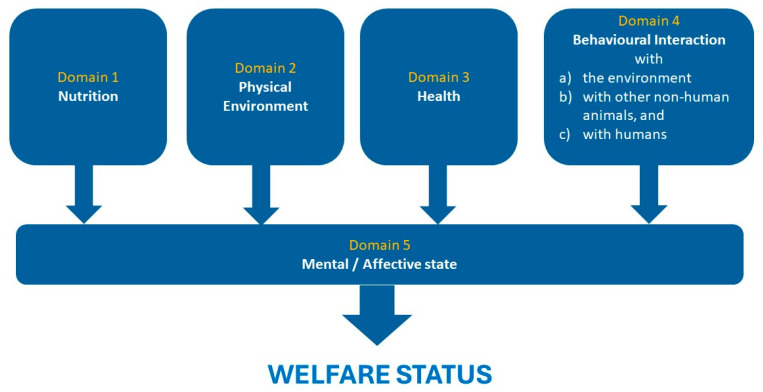
The Five Domains model adapted from Mellor et al. [21] (Creative Commons Attribution CC BY 4.0 DEED, https://creativecommons.org/licenses/by/4.0/ accessed on 13 July 2024).

**Figure 2 animals-14-02223-f002:**
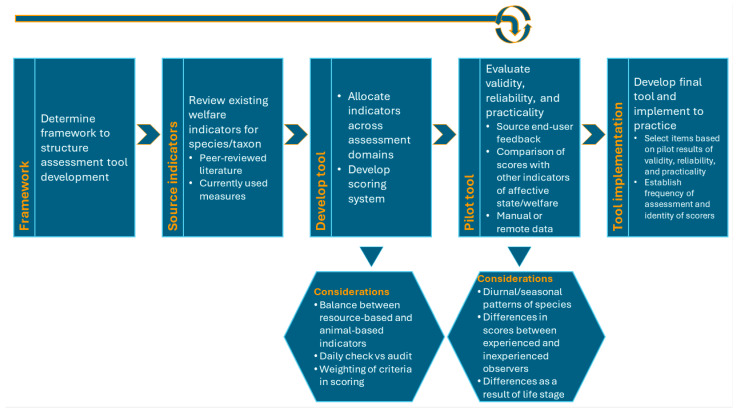
Modified from Jones et al. [88] (Creative Commons Attribution CC BY 4.0 DEED, https://creativecommons.org/licenses/by/4.0/ accessed on 13 July 2024), a flowchart outlining the process of developing a welfare assessment for use in zoos and the considerations at each step. This section discusses the process of development up to piloting, and, therefore, it does not cover tool implementation.

**Figure 3 animals-14-02223-f003:**
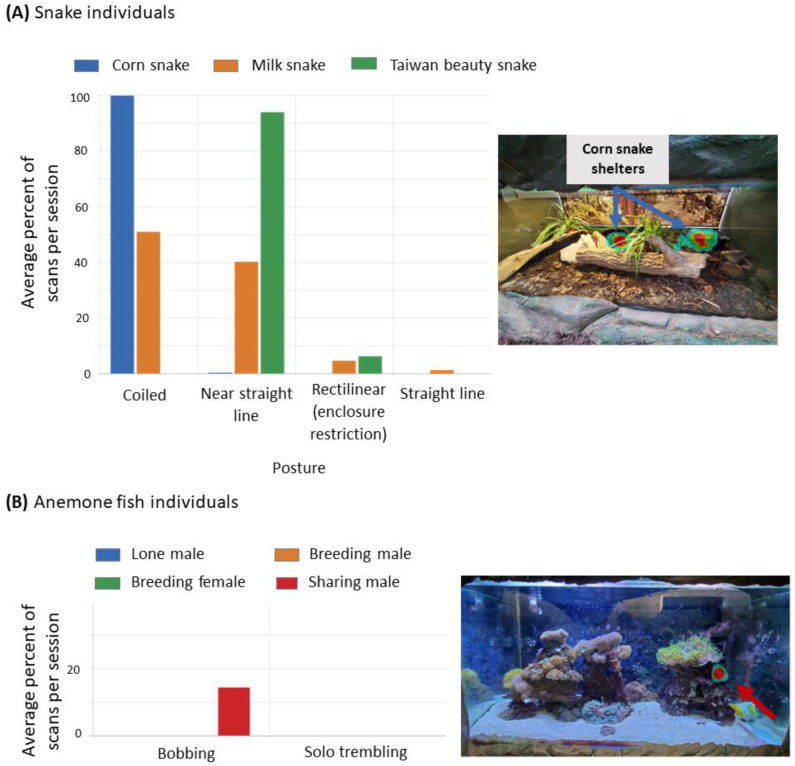
Data used to respond to behavioural category question, “No evidence of abnormal behaviour”, from the over-arching question, “Is a negative behaviour being displayed?”, for (**A**) snake individuals (11 sessions for corn snake, 1.83 h; 8 sessions for milk snake, 1.33 h; and 11 sessions for Taiwan beauty snake, 1.83 h) and (**B**) anemone fish individuals (11–12 sessions per individual, 1.83–2 h per individual). Graphs for both the snakes and anemone fish are time budgets created directly within the *ZooMonitor* app. These figures do not include measures of standard error. These results demonstrate that if figures for rhythms are too time-consuming to create, using data quickly taken directly from *ZooMonitor*, but analysed together can be immensely more informative than assessing only time budgets or only heat maps. The snake behaviours (**A**) are postures, with rectilinear being the only posture that is inherently negative, whereas the other postures are neutral. The enclosure map pictured for the snakes is of only the corn snake enclosure displaying enclosure use for all sessions and behaviours. The shelters indicated in the map are a bowl (left arrow) with an opening which allowed the snake to coil inside and a plastic box shelter (right arrow). For the anemone fish (**B**), the figure displays time budgets of two abnormal behaviours. In the enclosure map, the arrow is indicating where bobbing was displayed for the sharing male.

**Figure 4 animals-14-02223-f004:**
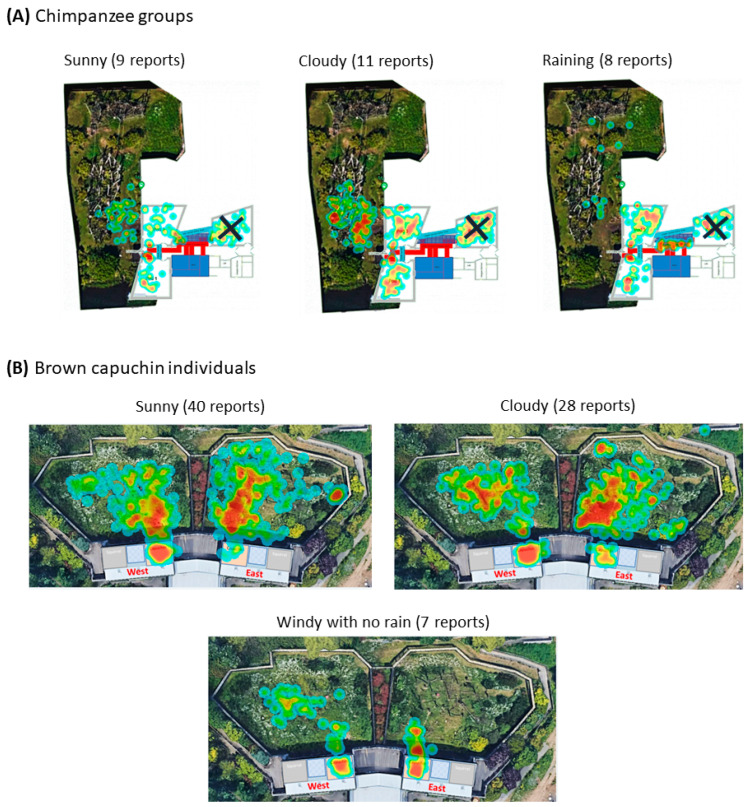
Data used to respond to the environment category question, “Has appropriate shade/shelter from weather/climate”, from the over-arching question “Is there appropriate shelter?”, for (**A**) chimpanzee groups and (**B**) brown capuchin individuals. The ability to create heat maps for specified weather conditions provides very useful information on what environmental features may be fit for different kinds of weather. This information could confirm that certain features are providing appropriate shelter or direct staff toward features that they may not realise could be enhanced to be fit for purpose. Each photo shows enclosure use for either both chimpanzee groups or all capuchin individuals during different weather conditions, with the number of sessions included in the heat map in parenthesis. For the chimpanzees, the black “X” indicates an enclosure pod that does not have access to the outdoors.

**Figure 5 animals-14-02223-f005:**
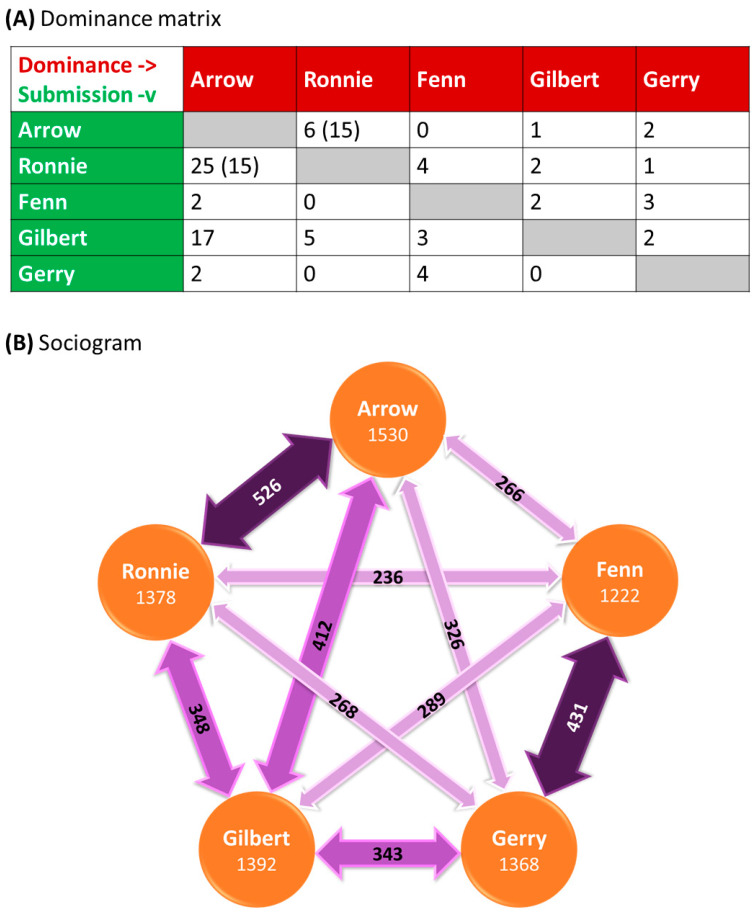
The (**A**) dominance hierarchy and (**B**) social dynamics of five giraffe bulls housed socially (28 sessions (560 interval data points) per individual equals 4.67 h per individual. In (**A**), we have a dominance matrix. It displays the number of times dominant and submissive behaviours were displayed (measured as all occurrence in *ZooMonitor*). Dominance behaviours include displace and bump, and submissive behaviour only includes submissive yielding. Numbers in brackets indicate the number of times the individuals were seen engaged in serious fights with one another with no clear winner or loser. The top row displays the giraffes when they were the actors of dominant behaviour or the recipients of submissive behaviour, and the first column displays the giraffes when they were submissive or the targets of dominance behaviour. In (**B**), we present a sociogram of the five male giraffes composed of recordings where giraffes engaged in social behaviour or spent in proximity with one another. “Social behaviour” includes “Approach”, “Follow”, “Necking”, and “Sparring”. Proximity data are gathered from all other behaviours and are included only from the focal animal, as it cannot be certain that two giraffes in proximity to the focal are necessarily in proximity to one another. Numbers below the giraffe’s name are the total number of interval data points where the giraffe was in proximity and/or displaying a social behaviour directed toward another giraffe. Numbers in the double-headed arrows are the total number of interval data points where the dyad members were socially interacting or in proximity of each other. The thickness and colour of the arrows also represent these numbers (light to dark colour/thinner to thicker arrow = fewer to more point samples). This figure and legend were created and written by the Master’s placement student (Callum Moore) who collected the data.

**Table 1 animals-14-02223-t001:** Summary of the strengths and weaknesses of the welfare assessment frameworks/processes considered for inclusion and exclusion in the welfare assessment tool developed. The checkmarks indicate which framework/process has the specified strength or weakness.

	Universal Animal Welfare Framework [89]	Evaluating Behaviour Budgets [92,93]	Animal Welfare Assessment Grid [90]	European Welfare Quality^®^ Project [91]	Sherwen et al. [94] Risk Assessment Based on Five Domains	24/7 across the Lifespan Approach [45]
Measures resource inputs	✓		✓	✓	✓	✓
Measures animal outputs		✓	✓	✓		✓
**Strengths**						
Identifies resource-based risks to welfare	✓		✓		✓	
Efficient	✓		✓		✓	
Non-invasive	✓	✓	✓	✓	✓	✓
Centralises animal in welfare assessment		✓	✓	✓		✓
Considers assessment of caretakers familiar with individual animals		✓	✓	✓		✓
Easily interpretable results	✓		✓		✓	
Addresses both positive and negative welfare states		✓		✓		✓
Addresses the continuous nature of welfare 24/7 across the lifespan						✓
**Weaknesses**						
Resource and labour intensive		✓		✓		✓
Does not evaluate animal-based indicators	✓				✓	
Results can be misleading due to lack of information on other domains of welfare	✓	✓			✓	
Mainly focuses on negative animal indicators			✓			

**Table 2 animals-14-02223-t002:** A list of the completed welfare assessments used to assess the validity and reliability of the RZSS Welfare Assessment Tool. Assessments were either on an individual or group and had 2–4 assessors completing the assessment questionnaire.

Species	Individual/Group	Number of Assessors
Chimpanzee	Group	3
Chimpanzee	Individual	3
Giant panda	Individual	3
Giant panda	Individual	3
Malayan Tapir	Individual	2
Mandrill	Individual	3
Mandrill	Group	3
Pelican	Group	3
Sulphur-crested cockatoo	Individual	4
Tegu	Individual	3
Yellow-breasted capuchin	Group	3
Yellow-breasted capuchin	Group	3
Yellow-breasted capuchin	Group	3
Yellow-breasted capuchin	Group	3
Amur tiger	Individual	3
Rhinoceros	Group	4
Yak	Individual	3

**Table 3 animals-14-02223-t003:** Concordance for responses to questions within each category.

Category	Question Range	# of Questions	Cronbach’s Alpha
Nutrition	1–4	4	0.7572
Health	5–13	9	0.6665
Environment	14–27	14	0.8934
Husbandry	28–36	9	0.8434
Behaviour	37–50	14	0.8822
Whole assessment	1–50	50	0.9318

**Table 5 animals-14-02223-t005:** Average of the overall Fleiss Kappas for all assessments. Under kind of assessments, “All” is the combination of the individual and group assessments. The number of assessments is the number of individual animals/groups assessed. However, each individual/group had multiple assessors, so the number in parenthesis is the total number of assessments completed by individual assessors. The average number matched is the number of responses matched by all assessors in a single assessment.

Kind of Assessments	Number of Assessments	Average Number of Assessors	Average Number Matched	Average Percent Matched	Fleiss Kappa
All	17 (52)	3.06	23.88	47.76	0.3483
Individual	9 (27)	3	24.33	48.67	0.3538
Group	8 (25)	3.13	23.38	46.75	0.3421

**Table 7 animals-14-02223-t007:** Details of the species for which behavioural observations were completed.

Species	Taxa	Individual/Group	Other Identifiers
Chimpanzee	Mammal	Group	2 groups separating mother and son through fission fusion groups
Yellow-breasted capuchin	Mammal	6 individuals	Female and male pairs in monkey house and 2 males housed individually
Brown capuchin	Mammal	6 individuals	3 males and 3 females; 3 from East group and 3 from West group; 2 from each rank (low, middle, and high)
Meerkat	Mammal	2 groups	Male group indoors, female group outdoors
Nubian giraffe	Mammal	5 individuals	5 males housed together
Eastern white pelican	Bird	7 individuals and group	3 male/female pairs and 1 male
Anemone fish	Fish	4 individuals	Mating pair and 2 neutral individuals
Corn snake	Reptile	Individual	Male
Milk snake	Reptile	Individual	Male
Taiwan beauty snake	Reptile	Individual	Male

**Table 8 animals-14-02223-t008:** Behavioural-domain questions and colour-coded grouping of them into over-arching questions being asked in the domain.

Colour-coding key for overarching questions	
	Is a natural behaviour being displayed at the appropriate levels?
	Is an interaction positive?
	Is a positive behaviour being displayed?
	Is a negative behaviour being displayed?
	**Behavioural-domain questions**
	Performs appropriate levels of self-care behaviours (grooming, preening, drinking, resting, and comfort activities)
	Has mostly positive interactions with conspecifics or other animals
	Has mostly positive or neutral interactions with staff/visitors
	Responds appropriately to novel changes in the environment (interest in appropriate enrichment vs. fear/aversion/apathy)
	Can express choice and control over being in different (indoor/outdoor) areas (except for maintenance periods)
	Exhibit appropriate territorial behaviour (patrolling, scent marking)
	Exhibits appropriate foraging and feeding behaviours
	Exhibits play behaviour (alone or socially)
	Exhibit appropriate levels of rest and sleep
	No evidence of dysfunctional social interactions
	No evidence of abnormal or stereotypic behaviour
	Exhibits reproductive behaviours as appropriate to the species and individual (courtship, mating, nest-building, incubating, birth, rearing, etc.)
	Exhibits species-specific behavioural needs (rooting, burrowing, climbing, perching, social grooming, etc.)

**Table 9 animals-14-02223-t009:** Select environmental-domain questions and colour-coded grouping of them into over-arching questions.

Colour-coding key for overarching questions		
		Are there appropriate shelters?
		Are the enclosure and furnishings allowing for natural behaviours to be displayed?
		**Select environmental domain questions**
		The size, shape and topography of the enclosure is appropriate for the species to exercise, explore, and exhibit normal territorial behaviours
		Substrates are suitable for the species (consider locomotion (abrasion, traction, and support); resting (comfort, depth, and cleanliness); foraging (depth and cleanliness); and burrowing (will support tunnels, depth, and secure))
		Has appropriate shelters, retreats, visual barriers, and off-show areas from conspecifics and visitors
		Has appropriate shade and shelter from weather/climate
		Planting is appropriate for the species, providing shelter, shade, retreats, microclimate provision, feeding and opportunities, and plants are not toxic and do not present an escape risk
		Furnishings allow appropriate species-specific behavioural needs (climbing, swinging, jumping, perching, nesting, stretching, hiding, sleeping, flight, etc.)

**Table 10 animals-14-02223-t010:** Session lengths, number of intervals, and number of sessions completed for each species’ *ZooMonitor* project.

Species	Type of Observation	Session Length	Session Intervals	Sessions Completed
Chimpanzee	Group	25 min	10	27 (11.25 h)
Eastern white pelican	Group	10 min	15	24 (4 h)
Meerkats	Group	10 min	10	37 (6.16 h)
Yellow-breasted capuchins	Individual	10 min	20	40 (6.67 h)
Brown capuchins	Individual	10 min	20	72 (12 h)
Nubian giraffe	Individual	10 min	20	140 (23.33 h)
Anemone fish	Individual	10 min	20	47 (7.83 h)
Snakes	Individual	10 min	20	30 (5 h)

**Table 11 animals-14-02223-t011:** Estimated time commitment for each stage of behavioural data collection. The 3-day minimum for stage 4 (data collection) is an estimate of the least amount of time it would take to complete 2 sessions per hour (a minimum standard) of the working day. Ideally, this would be repeated several times throughout the year (at least once per season), and the 3 days of collection would be spread across 2 weeks. *ZM* is *ZooMonitor*.

Data-Collection Stage	Time Commitment
Familiarisation with *ZooMonitor* and chosen animal ethogram	1 day
In-person practice using *ZooMonitor*	2 days
Reliability testing: (1) Ethogram quiz (80% to pass) (2) Video/in-person *ZooMonitor* reliability (70% reliability, 2–3 reliable sessions needed depending on difficulty of project)	Quiz: 1–3 days *ZM*: 2–5 days
Data collection for welfare assessment	3 days–1.5 weeks (dependent on species)

**Table 4 animals-14-02223-t004:** Questions that, when omitted, resulted in higher Cronbach’s alpha for their category. The omitted variable column indicates the question number in the assessment (Appendix A). The last column indicates the new Cronbach’s alpha for the category when the variable is omitted in comparison to the previous Cronbach’s alpha for the category seen in Table 3 (in brackets).

Omitted Variable	Category	Question	Cronbach’s Alpha When Omitted
2	Nutrition	A diet sheet is available for the species and is reviewed incorporating best practice guidelines for nutrition and evidence-based literature as available.	0.8732 [0.7572]
8	Health	Faeces are appropriately formed and normal for the species.	0.6823 [0.6665]
14	Environment	Temperature levels/gradients are within parameters appropriate for the species year-round (consider internal/external, seasonal, nighttime, appropriate variation, choice, records, etc.).	0.9069 [0.8934]
15	Environment	Humidity levels/gradients are within parameters appropriate for the species year-round (consider internal/external, seasonal, nighttime, appropriate variation, choice, records, etc.).	0.9073 [0.8934]
16	Environment	Light levels, quality and photo period are within parameters appropriate for the species year-round (consider UV, photoperiod, flicker/glare, colour, internal/external, seasonal, nighttime, appropriate variation, choice, records, etc.).	0.8956 [0.8934]
33	Husbandry	Water feature is safe and of a depth/size/volume/gradient that allows for species-specific natural behaviours.	0.8565 [0.8434]
45	Behaviour	Exhibits play behaviour (alone or socially).	0.8868 [0.8822]

**Table 6 animals-14-02223-t006:** Average Fleiss Kappa for each possible response within the assessment.

Kind of Assessments	No	Yes	Not Applicable	Unknown
All	0.2649	0.3829	0.6662	0.0925
Individual	0.1878	0.3803	0.7173	0.1320
Group	0.3517	0.3858	0.6087	0.0482

## Data Availability

The original contributions presented in the study are included in the article/Appendix A; further inquiries can be directed to the corresponding author. The data analysed for this study were from scheduled welfare assessments performed by RZSS Edinburgh Zoo. Therefore, privacy restrictions prevent us from making these data publicly available.

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
