# Peer review of "The Royal Zoological Society of Scotland’s Approach to Assessing and Promoting Animal Welfare in Collaboration with Universities"

_animals, 2024, doi:10.3390/ani14152223_

Round 1
Reviewer 1 Report
Comments and Suggestions for Authors
First of all, I want to congratulate all the authors of this paper for this very well-prepared manuscript and I'm very glad I accepted reviewing this draft, because it is a very interesting and important topic. I believe that this article, based on the topic as well as the quality, should be published in this journal. It fits all the journal criteria and even without undergoing any modification, in its current version, it would already be a fantastic addition for this journal, for the research community as well as the animal care community. Thus, all suggestions I'm going to be listing from now on are simple recommendations that can be introduced but do not have to be introduced if the authors do not believe them to lead to any improvement for this manuscript or the toolkit, considering that the toolkit is still in its development phase.
As the author mentioned in the beginning, animal welfare in captivity is becoming an increasingly popular research area, however, typically lacks validation and suggested methodologies and toolkits are often inefficient and not practical to be used and deployed in a variety of institutions and locations, or across species. I believe this project presented in this manuscript is a big step forward to achieving a toolkit and resources that can be very beneficial for many animal care institutions as well as the animals in their care.
Nevertheless, based on the current presented version of the toolkit, I believe there are several aspects that can still be improved. And I have no doubt that they will be improved as the authors say themselves, this is a work in progress, and they plan to keep developing and improving details and aspects.
The whole manuscript is very detailed providing much information about all the different steps, yet at some point it feels a little bit too long and extensive which might lead to the reader to lose focus. Especially the first section concentrating on all the research that has been done already is perhaps a little bit too extensive and takes away some of the focus that should fall more on the development and validation of the toolkit itself. Thus, I suggest shortening this part to emphasize more the toolkit development itself.
In the introduction part in several paragraphs, for example line 62 as well as line 103, you state that research is the driver for many conservation and welfare efforts. However, I suggest being careful with this kind of statement as research should be rather understood as extremely useful and even necessary tool to demonstrate and validate conservation efforts and its impact, yest research itself is not really the driver but simply a tool in our arsenal.
Regarding table one which basically lists research articles published referring to specific welfare areas is very impressive and interesting; however, it interrupts the reader's flow. Furthermore, as most if not all of these articles have been already mentioned and cited within the text anyway, I suggest moving this table to the supplementary material.
In the paragraph explaining the 4th domain (behavioral interaction), specifically interaction with humans (the lines 348 onwards) I was surprised to see that there is only one question related to the topic of human animal interaction and in this paragraph, the explanation feels more orientated towards the interaction between animals and staff members without giving much importance to the interaction between animals and visitors. After finishing the reading through the whole manuscript, I feel that this aspect might have received too little attention in this manuscript and the toolkit itself. As such I suggest in the further development to put more emphasis in this area and for this manuscript and publication just adding a brief explanation to this specific paragraph about the importance of considering human animal interaction also in terms of visitor activities
While reading the whole manuscript the first time, I made many annotations to provide feedback. However, to my very positive surprise, the further down I got, I could cross out these annotations one after another, because you kept on solving my doubts and suggestions yourself. As a reader, this tends to be a pleasurable experience.
Regarding the variance between assessors mentioned in line 652-656: Specifically, this part regarding the faeces makes me wonder if variance might have differed between group and individual evaluations. With the faeces example it is quite easy to imagine how one could evaluate the faeces of a group depending on the absence of any not ideal faeces, while another assessor might score this question simply based on the majority of faeces consistency. On the other hand, if we are focusing on one individual than there might be less variance. (Although faeces consistency might change even from one day to another and should best be evaluated by the care givers that regularly spot and clean these faeces).
When talking about the Fleiss Kappa results in line 669-671 the way how you present these results seems quite vague. By just using the words fair and strong, someone not familiar with these tests might overrate your results as fair (0.21-0.4) is actually 3 categories lower than strong (0.81-1.0). Thus, I suggest adding the threshold values to the words “fair” and “strong” for transparency, although this does not change anything in regard to the results or validity of this work.
Regarding the results of the different answers given and tested (mentioned in line 677): I am wondering when an assessor is supposed to leave a question in “blank”, considering that there is the option of “unknown”, which to me, seems to be the same answer. Thus, I would like to understand what the difference is in terms of information between these two answer possibilities and if it even makes sense to maintain and analyse them apart. If it is not needed to maintain them separately, the overall variance issues might improve. And in line 710, I am still surprised to see that both “yes” and “no” only achieved a “fair” agreement result, which makes me wonder if you compared (analysed separately) values also depending on species. I might imagine that some species might be perhaps more difficult to evaluate than others. Could it be that for example mammal species tend to reach a higher agreement score than for example reptiles or birds. Or do species that tend to consist of very few animals per group/zoo, such as pandas receive a higher agreement than species typically consisting of big groups such as a mandrill group for example? This is not at all meant as a critic or something that needs fixing, but rather an attempt to better understand these results and perhaps help identify solutions.
I am extremely pleased to see how the authors even go as far as to repeatedly include and emphasis the importance of cycles of behaviours (for example in line 732), which is something that too few consider in captive care and even less in research as it typically requires much more time investment and resources to cover this aspect. Authors especially emphasis differences between daytime, night-time and seasonality as well as take weather conditions into considerations. I suggest mentioning even more circles here in this manuscript (I am sure the authors are already aware of anyway) such as hormonal cycle, offspring related cycles, such as weaning in mammals, but also consider cycles within a day-to-day routine such as feeding and gating schedules.
In line 767 please make sure to cite the ZooMonitor app properly here in it´s first mention as well.
Regarding the behavioural data collection using ZooMonitor: Around line 822 I suggest it would be necessary and useful for the reader to explain which data collection methods you were using for the different species. This bit might be kept quite brief and refer to a table in the supplementary material that provides more details. From what I can see later on, you mostly used interval sampling method and some all-occurrence sampling as well. By providing this information at this point, understanding the use and data collection you mention in upcoming examples might make a lot more sense to the readers and would be easier to follow.
Regarding the data collection methodology, you state in line 897, that “This information shows how sessions do not need to be long to gain decent estimates of behaviour for groups and individuals.” I absolutely do agree that session duration do not need or should not be too long, however, the shorter the sessions the more frequent they should be conducted in order to really objectively reflect the animal’s behavioural state.
Regarding the observer training mentioned in lines 941-944: The fact that not all observes eventually passed the reliability tests also indicates that your procedures and standards, values quality and precision over quantity (which is something positive). Although it is a pity to lose a potential observer especially if they are motivated, it may be more important and admirable to use a method were data quality remains high, as you try to achieve.
Furthermore, in line 945 you state “With university students, the latter may not be possible as the students may be required to produce theses or reports.” The fact that they do not manage to pass the reliability testing simply means that you can´t use their data nor they can use the data of other observers, yet they could still make use of their own data (isolated) for a university work.
Regarding the questions listed in the supplementary material, other than just providing the questions themselves, it would be helpful if you could also provide the instructions (rule book) an assessor would receive to fill these questions out.
Overall, this is a fantastic manuscript worth being published in this journal and even without modifying any of the suggestions listed beforehand, it already reaches the quality and importance necessary to be published here. Once again, I want to congratulate the authors, and everyone involved in this project for a remarkably interesting manuscript. And I am very much looking forward to seeing the final future versions of this toolkit.
-
-
Reviewer 2 Report
Comments and Suggestions for Authors
The topic is highly relevant as it addresses a critical aspect of animal welfare in zoos, which is a growing concern globally. The manuscript, although extensive and perhaps somewhat tiring to read, provides an excellent background and delivers on the proposed theme. It is evident that the majority of the co-authors have significant expertise in this area.
1. Brief Summary
The paper aims to describe the approach of the Royal Zoological Society of Scotland (RZSS) in assessing and promoting animal welfare at Edinburgh Zoo and Highland Wildlife Park in collaboration with universities. The main contributions include the development and validation of the RZSS Welfare Assessment Tool and the Stirling Toolkit, which provide standardized, evidence-based methods for evaluating animal welfare. The strengths of the paper lie in its comprehensive methodology, collaborative approach, and the practical application of welfare science to improve animal care in zoological settings.
2. General Concept Comments
The methodology is highly detailed and covers several important aspects of animal welfare assessment, including the development of a welfare assessment tool and the validation of questions and reliability of responses. The results presented are consistent with the described methodology. Validity and reliability analyses were conducted as described, and the results are clearly presented. The discussion adequately encompasses the findings and offers practical suggestions for improving the assessment tool and data collection. The conclusion is factual and reflects the study's findings, highlighting the importance of a comprehensive and collaborative approach to assessing and promoting animal welfare.
While the methodology is robust, the paper could benefit from a more detailed discussion on the limitations of the current tools and how they might be addressed in future research. Additionally, the paper could explore more deeply the potential challenges in implementing these tools across different zoological institutions with varying resources.
3. Specific Comments
Line 50-51: Clarify the statement regarding the lack of extensive research on the ability of many species to experience pain and their natural behaviors. Provide specific examples or references to support this claim.
Line 70: Divide into a new paragraph.
Line 137-148: Consider incorporating the One Conservation concept in addition to the One Plan Approach, including a paragraph on this concept. One Conservation is a concept focused on connecting conservation actors and seeks to "bridge the gap between wild and captive population management," understanding that it centers on parties already involved in the conservation of endangered species (wildlife conservationists and the zoo/aquarium community). The One Conservation concept seeks to show that it would be important to change paradigms of other parties never engaged in conservation programs but are fundamental to ensuring not only the conservation of species but also of ecosystems. Being considered in its definition as "an interconnection between ex situ and in situ conservation plans, anthropic actions on the environment (sustainability), and research in different areas that encompass conservation," this interconnection is discussed in greater depth in the integrated vision part of the article, where we point out the current gap between agribusiness and the conservation community. Suggested references to understand this concept:
https://doi.org/10.1590/1984-3143-AR2021-0024
https://doi.org/10.3389/fvets.2022.897404 (about the need for protocols)
https://doi.org/10.1016/j.therwi.2023.100024
4. Final Reviewer Comments
This work can significantly contribute to improving animal welfare in zoos worldwide by providing standardized, evidence-based tools for animal welfare assessment. The RZSS Welfare Assessment Tool and the Stirling Toolkit can be adopted by other institutions to ensure that animals receive consistent and high-quality care. Additionally, collaboration with universities can encourage more research and the collection of detailed behavioral data, allowing for more effective and personalized interventions. The dissemination of these tools and practices can promote a more holistic and scientific approach to animal welfare management, benefiting both the animals and the institutions that house them. For these reasons, incorporating the One Conservation vision could add value to the manuscript.
This reviewer congratulates the authors on this manuscript.
While not necessary for this manuscript, the following references might be useful for future work:
https://doi.org/10.3390/ani13081277
https://doi.org/10.1016/j.biocon.2023.110345
https://doi.org/10.1017/S0962728600000543
Comments on the Quality of English LanguageThe quality of English in the manuscript is high, with good structure, correct grammar, and appropriate use of technical terminology. However, there are some areas where the complexity of sentences can be simplified to improve readability. For example, some sentences are long and can be divided into shorter sentences to facilitate understanding.
